# A Hybrid Variational-Ensemble data assimilation scheme with systematic error correction for limited area ocean models

Paolo Oddo[1], Andrea Storto[2], Srdjan Dobricic[3], Aniello Russo[1], Craig Lewis[1], Reiner Onken[1], Emanuel Coelho[1]

[1]NATO Science and Technology Organization, Centre for Maritime Research and Experimentation, Viale San Bartolomeo 400, 19126 La Spezia, Italy
[2]Euro-Mediterranean Centre for Climate Change (CMCC), via Franceschini 31, I-40128 Bologna - Italy
[3]European Commission, Joint Research Centre, Via Enrico Fermi 2749, I - 21027 Ispra (VA), Italy

*Correspondence to:* Paolo Oddo (Paolo.Oddo@cmre.nato.int)

**Abstract.** A hybrid variational-ensemble data assimilation scheme to estimate the vertical and horizontal parts of the background-error covariance matrix for an ocean variational data assimilation system is presented and tested in a limited area ocean model implemented in the western Mediterranean Sea. An extensive dataset collected during the Recognized Environmental Picture Experiments conducted in June 2014 by the Centre for Maritime Research and Experimentation has been used for assimilation and validation. The hybrid scheme is used to both correct the systematic error introduced in the system from the external forcing (initialization, lateral and surface open boundary conditions) and model parameterization and improve the representation of small scale errors in the Background Error Covariance matrix. An ensemble system is run off-line for further use in the hybrid scheme, generated through perturbation of assimilated observations. Results of four different experiments have been compared. The reference experiment uses the classical stationary formulation of the background error covariance matrix and has no systematic error correction. The other three experiments account, or not, for systematic error correction and hybrid background error covariance matrix combining the static and the ensemble derived errors of the day. Results show that the hybrid scheme when used in conjunction with the systematic error correction reduces the mean absolute error of temperature and salinity misfit by 55% and 42% respectively versus statistics arising from standard climatological covariances without systematic error correction.

## 1 Introduction

The study and the characterization of the ocean is a complex discipline involving different aspects of modern science. In order to obtain a coherent and time evolving three-dimensional picture of the ocean from historical and present day observations, and be able to predict the future evolution of the environment we need to solve theoretical and technical issues.

It is not feasible to observe all variables of interest with adequate spatial and temporal scales. Modern technologies, like satellite remote sensing and autonomous vehicles, have significantly increased our capability to observe the environment in general and the ocean in particular. However, the huge number of degrees of freedom characterizing the ocean state still prevents sampling at the desired resolution. In order to fill the observational gaps and expand the temporal horizon covered by the observations (both in the past and in the future), oceanographers combine direct observations with theoretical studies by means of models and data assimilation.

A numerical hydrodynamic model is basically the discretized version of the Primitive Equations, it is an approximation of nature. Moving from the continuous to the discrete space, additional approximations are introduced and should be accounted for when analysing model results. These approximations affect the model solutions in terms of quality and accuracy and, more importantly, differences between the numerical solution and the true state amplify along time due to the chaotic component of the ocean dynamic.

In order to minimize these differences and improve the quality and accuracy of model results, data assimilation techniques have been developed during the past decades. Data assimilation is a technique to correct the model solution based on statistical and physical constraints derived from observations and model simulations.

Even if different kinds of data assimilation techniques exist, most of them rely on the same basic principle, the combination of physically based and statistical approaches to maximize the conditional probability of the model state given the observation. Data assimilation schemes developed for oceanographic studies can be classified in two categories. The first one is the Kalman filter (KF) type algorithms with Background Error Covariances (BECs) matrices usually derived from ensemble statistics (Evensen, 2003). The second type of assimilation algorithms employ stationary BECs derived from long-term model integrations (Yin et al. 2011; Weaver and Courtier 2001; Pannekoucke and Massart 2008). A key avenue to improving data assimilation is accurate specification of the error statistics for the background forecast, also known as the prior or first guess (Schlatter et al. 1999).

The Ensemble Kalman Filter (EnKF) (Evensen, 1994) consists of a set of short term forecasts and data assimilation cycles. In the EnKF, the BECs are estimated from an ensemble of model simulations. The presumed benefit of utilizing these ensemble-based techniques is their ability to provide a flow-dependent estimate of the BECs. The traditional EnKF incorporates probabilistic information on analysis errors in the generation of the ensemble by imposing a set of perturbations for each ensemble member, generating the individual numerical forecasts from different sets of initial conditions implied by the different sets of observations and/or different numerical model configurations. The EnKF is related to the classic Kalman filter (KF), which provides the optimal analysis in the case that the forecast dynamics are linear and both background and observation errors have normal distributions. The main difference is that the KF explicitly forecasts the evolution of the complete forecast error covariance matrix using linear dynamics, while the EnKF estimates this matrix from a sample ensemble of fully nonlinear forecasts. The EnKF also addresses the computational difficulty of propagating or even storing the forecast error covariance matrix. Using ensemble simulations implies also that EnKF does not assume the covariances to propagate linearly.

On the other hand, many current and past operational data assimilation methods use long time series of previous forecasts to develop stationary and often also spatially homogeneous approximations to BECs. Schemes that use such statistics include optimum interpolation and three-dimensional variational data assimilation (3DVAR), and have the advantage of being less computationally demanding, namely allowing higher resolution. In reality, BECs may vary substantially depending on the flow and error of the day. A four-dimensional variational data assimilation (4DVAR) system implicitly includes a time-evolving covariance model through the evolution of initial errors under tangent linear dynamics (Lorenc 2003) within the assimilation time window. However, the time evolving covariance model may still be limited by usage of a stationary covariance model at the beginning of each 4DVAR cycle. Furthermore, like the EnKF, 4DVAR is computationally intensive, requiring multiple integrations of tangent-linear and adjoint versions of the forecast model. The specification of flow-dependent statistics is per se a demanding task, due to the difficulty of retrieving information on errors in model space.

The ensemble EnKF provides an alternative to variational data assimilation systems. Under assumptions of linearity of error growth and normality of observation and forecast errors, it has been proved that the EnKF scheme produces the correct BECs as the ensemble size increases (Burgers et al. 1998). However for smaller ensembles, the EnKF is rank deficient and its BEC estimates suffer from a variety of sampling errors, including spurious correlations between widely separated locations that need to be removed by means of specific techniques (e.g. covariance filtering or localization).

Assimilation methods using a static type of the BEC have recently gained considerable attention because of their flexibility (Lorenc 2003). Furthermore present computational resources limit the number of ensemble members accounted on operational EnKF. Thus, it is appealing to have an algorithm that could work with smaller-sized ensembles and that could benefit from whatever flow-dependent information this smaller ensemble provides.

Recent encouraging results suggest that if ensemble information is used in the variational data assimilation framework to augment the static BEC, analyses can be improved. Hereinafter, we call this method a "hybrid" scheme. Development of hybrid schemes has been an area of active research in atmospheric data assimilation (Hamill and Snyder 2000; Etherton and Bishop 2004; Wang et al. 2007). Several studies have been conducted on the hybrid schemes. Studies by Hamill and Snyder (2000), Etherton and Bishop (2004), and Wang et al. (2007) used simple models and simulated observations to suggest the effectiveness of incorporating ensembles in the 3DVAR to improve the analyses. Although the recourse to hybrid covariances and the choice of the relative weights given to them remains empirical in practice, it has been shown in particular that hybrid models tend to be more robust than conventional ensemble based data assimilation schemes, especially when the model errors are larger than observational ones (Wang et al. 2007, 2008, 2009). This feature is attractive for the regional assimilation problems in oceanography, where information on the background state is often scant and incomplete. Promising application of the hybrid scheme in global oceanographic exercise has been recently provided by Penny et al. (2015). They compared hybrid, classical 3DVAR and EnKF schemes in an observing system simulation experiment and using also real data, showing that the hybrid scheme reduces errors for all prognostic model variables eliminating growth in biases present in the EnKF and 3DVAR.

Recent works have also started addressing the issue of multi-scale data assimilation, where the analyses are combination of corrections with different spatial scale signals, assuming somehow that spatial scale are separable and that observations may naturally bear information across several spatial scales. Examples of these schemes range from multi-scale 3DVAR systems (MS-VAR), sequential applications of horizontal operators with different correlation length-scales (Mirouze et al., 2016), or

inclusion of a large-scale analysis in the analysis formulation as additional constraint (Guidard and Fischer 2008). A possible simplification is to assume that systematic errors are characterized by long length scales, as often occurs to some extent (Dee 2005).

In this study, we develop a hybrid data assimilation system for the REP14-MED (Mediterranean Recognized Environmental Picture 2014) NEMO model implementation, based on the existing 3DVAR system. Section 2 describes the hybrid variational

data assimilation scheme adopted accounting for systematic error corrections. In Section 3 details on the experiments set-up are provided. In Section 4 the results are presented and discussed. Finally Section 5 offer the summery and conclusions.

## 2 The Hybrid Variational-Ensemble Scheme

A 3DVAR algorithm has been used to implement and test our hybrid assimilation scheme. 3DVAR is relatively easy to implement and to expand, it can easily take into consideration different estimates of BEC, its core is independent of the

primitive equations model core, and it is portable. The cost function in 3DVAR is defined as:

$$J(\boldsymbol{x}) = \frac{1}{2}(\boldsymbol{x} - \boldsymbol{x_b})^T \mathbf{B}^{-1}(\boldsymbol{x} - \boldsymbol{x_b}) + \frac{1}{2}(\boldsymbol{y} - H[\boldsymbol{x}])^T \mathbf{R}^{-1}(\boldsymbol{y} - H[\boldsymbol{x}]) \tag{1}$$

where, $\boldsymbol{x}$ is the analysis state vector at the minimum of $J$, $\boldsymbol{x_b}$ is the background state vector, $\mathbf{B}$ is the background error covariance

matrix, $H$ is the non-linear observational operator, $\boldsymbol{y}$ are the observations and $\mathbf{R}$ is the observational error covariance matrix. The cost function is linearized around the background state and take the following form:

$$J(\delta\boldsymbol{x}) = \frac{1}{2}\delta\boldsymbol{x}^T \mathbf{B}^{-1}\delta\boldsymbol{x} + \frac{1}{2}(\mathbf{H}\delta\boldsymbol{x} - \boldsymbol{d})^T \mathbf{R}^{-1}(\mathbf{H}\delta\boldsymbol{x} - \boldsymbol{d}) \tag{2}$$

where $\boldsymbol{d} = [\boldsymbol{y} - H(\boldsymbol{x_b})]$ is the misfit, $\mathbf{H}$ is the linearized observational operator and $\delta\boldsymbol{x} = \boldsymbol{x} - \boldsymbol{x_b}$ are the increments. Following Dobricic and Pinardi (2008), the present 3DVar scheme assumes that the $\mathbf{B}$ matrix can be rewritten and thus decomposed as follows:

$$\mathbf{B} = \mathbf{V}\mathbf{V}^T \tag{3}$$

$$\mathbf{V} = \mathbf{V_D}\mathbf{V_{u,v}}\mathbf{V_\eta}\mathbf{V_H}\mathbf{V_V}. \tag{4}$$

This also has the advantage of imposing pre-conditioning, as the minimization is performed on the control variable $v$ (with $\delta x = \mathbf{V}v$), which also serves the purpose of avoiding the inversion of $\mathbf{B}$.

Basically, the background error covariance matrix is modeled as a linear sequence of several $\mathbf{V}$ operators. Each $\mathbf{V}$ defines a specific error space. From right to left $\mathbf{V}_v$ defines the vertical covariance computed using multivariate Empirical Orthogonal Functions, $\mathbf{V}_H$ projects the vertical error to the horizontal space by means of a recursive filter, $\mathbf{V}_\eta$ (the balance operator) is a 2D barotropic model accounting for sea surface height adjustments and $\mathbf{V}_{u,v}$ force a geostrophic balance between temperature, salinity and the velocity components. Finally, $\mathbf{V}_D$ is a divergence damping operator avoiding spurious currents close to the coast in the presence of complex coast lines (details in Dobricic and Pinardi 2008). It is clear that this $\mathbf{B}$ formulation introduces flexibility in the code and allows the possibility to test different hypotheses.

In our static formulation of the 3DVAR, the vertical transformation operator $\mathbf{V}_v$ has the form:

$$\mathbf{V_V} = \mathbf{S_c}\boldsymbol{\Lambda_c}^{1/2} \tag{5}$$

where columns of $\mathbf{S_c}$ contain multivariate eigenvectors and $\boldsymbol{\Lambda_c}$ is a diagonal matrix with eigenvalues of EOFs. Promising recently published results (Dobricic et al. 2015) propose a new method to estimate the vertical part of the background-error covariance matrix for an ocean variational data assimilation system based on high frequency estimates from a Bayesian Hierarchical Model. A general approach in defining hybrid assimilation schemes is to compute $\mathbf{B}$ as a linear combination of the "static" covariance operator, $\mathbf{B_c}$ and the flow-dependent operator, $\mathbf{B_e}$, derived from the statistics of an ensemble of analyses/forecast:

$$\mathbf{B} = \alpha\mathbf{B_c} + (1-\alpha)\mathbf{B_e} \tag{6}$$

The relative weighting ($\alpha$) still requires empirical tuning but in general can be adjusted to the size of the ensemble. Large ensemble size can provide robust estimate of $\mathbf{B_e}$ and thus Eq. (6) can be theoretically implemented with small $\alpha$ values (Menetrier and Auligne', 2015).

The proposed approach introduces the flow-dependent $\mathbf{B}$ by defining the increment as a weighted sum of parts corresponding to climatological and ensemble covariance matrices:

$$\delta x = \delta x_c + \delta x_e. \tag{7}$$

It can be demonstrated (see Appendix for details) by combining Eq. (2), Eq. (6) and Eq. (7) that the following cost function has the minimum for the same value of $\delta x$ as the cost function with the background-error covariance matrix defined in Eq. (6) ( e.g. Wang et al. 2007):

$$J(\delta x) = \frac{1}{2}\delta x_c^{T}(\alpha \mathbf{B}_c)^{-1}\delta x_c + \frac{1}{2}\delta x_e^{T}((1-\alpha)\mathbf{B}_e)^{-1}\delta x_e + \frac{1}{2}(\mathbf{H}\delta x - \mathbf{d})^T\mathbf{R}^{-1}(\mathbf{H}\delta x - \mathbf{d}) \tag{8}$$

In Appendix we further demonstrate that by updating each member of the forecast ensemble by Eq. (8) we obtain the same estimate for the analysis error covariance matrix as when doing it with Eq. (6).

By defining the control vector $v$ consisting of climatological and ensemble parts $v = (v_c, v_e)$ the cost function becomes:

$$J(v) = \frac{1}{2}v_c^{T}v_c + \frac{1}{2}v_e^{T}v_e + \frac{1}{2}(\mathbf{H}\delta x - \mathbf{d})^T\mathbf{R}^{-1}(\mathbf{H}\delta x - \mathbf{d}), \tag{9}$$

and increment $\delta \mathbf{x}$:

$$\delta x = \left(\mathbf{V}_D\mathbf{V}_{u,v}\mathbf{V}_\eta\mathbf{V}_H\right)\left(\alpha^{1/2}\mathbf{S}_c\Lambda_c^{1/2}v_c + (1-\alpha)^{1/2}\mathbf{S}_e\Lambda_e^{1/2}v_e\right), \tag{10}$$

where columns of $\mathbf{S}_e$ and $\Lambda_e$ can now be computed at any frequency from a relatively small size ensemble.

Ensemble statistics may also provide estimates of the day of the horizontal correlation radii to be used in $\mathbf{V}_H$. Using the recursive filter formulation $\mathbf{V}_H$ takes the form:

$$\mathbf{V}_H = \mathbf{W}_y(L_y^\varepsilon, \Delta y)\mathbf{G}_y(L_y^\varepsilon, \Delta y)\mathbf{W}_x(L_x^\varepsilon, \Delta x)\mathbf{G}_x(L_x^\varepsilon, \Delta x) \tag{11}$$

where $\mathbf{G}_x$ and $\mathbf{G}_y$ represent the zonal and meridional recursive filter operators, $\mathbf{W}_x$ and $\mathbf{W}_y$ are the diagonal matrices with normalization coefficient, $L_{x,y}^\varepsilon$ and $\Delta x,y$ are the zonal and meridional length scale and grid spacing respectively.

25  According to Belo Pereira and Berre (2006), for any simulated error $\varepsilon$ it is possible to define a zonal and a meridional length scales:

$$L_{x,y}^\varepsilon = \sqrt{\frac{\sigma^2(\varepsilon)}{\sigma^2\left(\frac{\partial \varepsilon}{\partial x,y}\right) + \left(\frac{\partial \sigma(\varepsilon)}{\partial x,y}\right)^2}} \tag{12}$$

30  where $\partial x,y$ are the derivatives in $x$ and $y$ direction, $\sigma^2(\varepsilon)$ and $\sigma^2(\partial \varepsilon/\partial x,y)$ are the variances of the background error and of the derivative. In the ensemble based approach $\varepsilon$ are the ensemble anomalies computed with respect to the ensemble mean.

Though most data assimilation methods assume that the model forecast (i.e. the background) is unbiased, that is rarely the case. Model bias can systematically cause the model to drift away from the truth, eventually propagating into the analyses. In Limited Area Models (LAM) integrated for relatively short time the systematic errors (bias) may derive from inadequate model physics and parameterizations as well as inaccurate initialization and open boundary conditions, including the atmospheric forcing. An adequate solution is strictly necessary since the systematic error in the large scale forcing field can prevent the right small scale dynamics from developing properly and thus can strongly reduce the potential benefits deriving from the increased resolution and/or improved physics.

Here, we assume that systematic errors are associated to large-scale errors. This idea is consistent with the high-resolution model presented in Section 3 and with the experimental setup where the large scale uncertainties (initialization, boundary conditions and surface forcing) are not accounted in the generation of the ensemble members.

Further expanding the decomposition introduced in Eq. (7) and following recent studies suggesting the possibility to treat multiple scales errors during the analysis steps (Wang et al. 2014; Li et al. 2015) we reformulate the analysis increments as:

$$\delta x = \delta x_c + \delta x_e + \delta x_s . \tag{13}$$

where the first two terms on r.h.s represent the increments deriving from the minimization of Eq. (9) while the last term indicates the increments due to the large scale systematic error not sampled either in the climatological or ensemble based estimates of $\mathbf{B}$. Note that the scale decomposition requires that large-scale and small scale background error are mutually uncorrelated. It is worth mentioning that the large scale systematic error could be partially accounted in the generation of the ensemble members, however this would imply a considerably large number of ensemble members with clear implications on the computational side and the corresponding $\mathbf{B_e}$ would then incorporate error information at different scales.

The availability of an ensemble simulation allow us to retrieve estimates of the model bias or systematic error. Recalling that:

$$\boldsymbol{d} = [\boldsymbol{y} - H(\boldsymbol{x_b})] = \boldsymbol{y} - H(\boldsymbol{x_t}) + H(\boldsymbol{x_t}) - H(\boldsymbol{x_b}) = \varepsilon_o - H(\varepsilon_r + \varepsilon_s) \tag{14}$$

where $\boldsymbol{d}$ is the misfit, $\boldsymbol{x_t}$ is the true state of the ocean, $\boldsymbol{\varepsilon_o}$ is the observational error resulting from the sum of $\boldsymbol{\varepsilon_{oi}}$ and $\boldsymbol{\varepsilon_{or}}$ (the instrumental and representation observational errors respectively), $\boldsymbol{\varepsilon_r}$ is the background random error and $\boldsymbol{\varepsilon_s}$ is the background systematic error (errors are defined as departures from the true state). Every assimilation scheme is designed to correct the random error which is assumed to have zero mean. The representation error can be defined based on the knowledge of the dynamics of the simulated system or as being proportional to the variance of the measurements (Oke and Sakov 2010). $\mathbf{S_{e,c}}$ and $\mathbf{\Lambda_{e,c}}$ introduced in the vertical covariance (Eq.10) provide a multivariate statistical representation of $\boldsymbol{\varepsilon_r}$. To obtain a bias, or systematic error, estimate we can average over the ensemble members and assuming that also the observational error is unbiased:

$$\overline{d} = -\overline{\varepsilon}_s \tag{15}$$

since also $\overline{\varepsilon}_r$ is zero by definition. Thus, analyzing the misfit of the ensemble members, we can obtain an estimate of the bias or systematic error. In other words, the ensemble system is exploited not only to estimate the flow-dependent components of the background-error covariances, but also to estimate the large-scale bias in the analysis step. From the previous relationship it is clear that the large-scale bias is originally defined in observation space and successively mapped in model space.

In our formulation we assume that the scales in $\varepsilon_s$ and $\varepsilon_r$ are significantly different and the estimate of the ensemble systematic error is used simultaneously to the 3DVAR analysis step to correct the background fields. The small scale increments arise from the classical minimization of the cost function $J$:

$$\delta x_c + \delta x_e = min_{\delta x}(J(\delta x)) \tag{16}$$

while the large scale increments due to the systematic error are defined as:

$$\delta x_s = L\left(\left(\overline{x_{eb}} + \mathbf{P}^T(\bar{\mathbf{d}})\right) - x_b\right) \tag{17}$$

where $\overline{x_{eb}}$ is the background ensemble mean, $L$ is a low pass filter used to ensure that scales of the two increments do not overlap and $\mathbf{P}$ is a generic linearized observational operator.

Such a scheme thus requires a fairly dense observational network to estimate the bias, whose availability may in general depend on the simulation area and period. The method is potentially affected by systematic observational errors and thus is sensitive to the design of the observational networks. On the other hand the analysis of the systematic error can provide useful insights about the error not represented in the ensemble space and thus help in the definition of the ensemble generation procedure. Depending on data availability and ensemble size the bias estimator can be constant or spatially or temporal dependent.

## 3 Experimental set-up

In June 2014, a Recognized Environmental Picture (REP14-MED) sea trial off the west coast of Sardinia was conducted by CMRE (Centre for Maritime Research and Experimentation), coordinating efforts of 20 partners from six different nations. Two research vessels collected a massive amount of data in an area of approximatively 100 km x 100 km with various oceanographic instruments (lowered CTD, undulating towed vehicle, CTD chain, ship mounted ADCP, shallow and deep underwater gliders, moorings, surface drifters and profiling floats). Complementary information has been retrieved, by remote sensing (sea level anomalies and sea surface temperature both from Copernicus Marine Environmental Monitoring Service, marine.copernicus.eu) and existing results from atmospheric and oceanographic operational models. *In-situ* CTD and glider

data together with remotely sensed SLA (Sea Level Anomalies) and SST (Seas Surface Temperature) have been used for assimilation while for the sake of simplicity only CTD data are used for model validations. Spatial and temporal distributions of observations are provided in Fig.1 and Fig.2, respectively.

NEMO (Nucleus for European Modelling of the Ocean, Madec 2008) has been implemented as the Primitive Equations dynamical model component of the data assimilation system. The ocean engine of NEMO is adapted to regional and global ocean circulation problems. Prognostic variables are the meridional and zonal velocities, sea surface height, temperature and salinity. In the horizontal direction, the model uses a curvilinear orthogonal grid and in the vertical direction, a full or partial step z-coordinate, or s-coordinate, or a mixture of the two can be applied. The NEMO ocean engine is very flexible allowing several choices for discretization and parameterizations; details on the present configuration are provided on Table 1 while model domain and bathymetry are shown in Fig.1.

An ensemble of data assimilation system with fourteen (14) independent members with daily assimilation cycles has been performed to generate the ensemble statistics. All simulation/assimilation experiments presented hereafter started on 1 June and ended on 30 June 2014, the MED-REP14 period. All the experiments are initialized and forced at the lateral open boundaries using the Mercator-Ocean (Drévillon et al. 2008) product in the Mediterranean Sea, while surface fluxes are computed by means of bulk formulae using hourly atmospheric data with 7.0 km horizontal resolution provided by the Italian Meteorological Centre and based on COSMO-ME model, an implementation of the Consortium for Small scale Modelling (COSMO). The ensemble members have been generated simultaneously assimilating perturbed observations varying the corresponding observational error, and assuming different horizontal correlation radii in $V_H$.

For the observation perturbation, either weak or strong criteria for retaining observations are used among the ensemble members. Conservative quality check procedures assume good quality flags in both temperature and salinity and reduce the total number of assimilated observations. Filters have been applied horizontally and vertically to reduce the higher spatial sampling of observation with respect to the model grid. Within the ensemble members, different vertical cut-off scales have been used in the low pass filter resulting in differently smoothed profiles. Horizontal data binning has been applied to the observations falling in 1 or 2 model grid cells while keeping the original vertical resolution. When the filtering or binning procedures are applied the corresponding full resolution profile standard deviation has been used as an estimate of the observational error. Similar procedures have been applied to CTD and Gliders data.

The default horizontal correlation radii ($L_{x,y}^\varepsilon$) have been computed according to Eq. (12) from 15 years CMEMS Mediterranean reanalysis. They correspond to 21 km and 12 km in the meridional and zonal directions respectively. Two additional sets of correlation radii have been used in the ensemble generation, they have been defined on the base of sensitivity experiments and correspond to 12/6 (meridional/zonal) and 6/3km respectively. The three sets of correlation radii remain constant during the simulated period. An example of the observational perturbation, associated error and horizontal correlation radii is shown in Fig.3.

The ensemble generation method spans the uncertainty linked with the observational sampling and assimilation formulation, implicitly acting on the background ensemble spread. This method clearly connects the growth of the ensemble spread to the

function used to perturb the observations and simultaneously links the ensemble spread to observation availability. For the time being, the perturbation of surface and lateral boundary conditions is not considered, assuming that the flow-dependent component of **B** is associated with the small-scale error fluctuations that are reasonably well reproduced with the observation perturbation only. This approach ensures also consistency with the assumptions done to derive Eq. (8). Although the large-scale forcing may act as an attractor for the ensemble perturbations, especially at the sea surface and in proximity of the boundaries, the goal of the present implementation is the evaluation of the feasibility of a hybrid system that simultaneously correct the model systematic and random errors. Further refinements of the ensemble generation strategy may be considered in the future.

All the ensemble members use a static and homogeneous **B** where $\mathbf{S}_c$, and $\mathbf{\Lambda}_c$ derive from multivariate (temperature, salinity and sea surface height) vertical EOF computed from the anomalies with respect to the long-term mean of a 15-year CMEMS Mediterranean reanalysis. The Incremental Analysis Update (IAU) strategy has been used to incorporate analysis increments into the model integration in a gradual manner (Bloom et al. 1995), spreading the analysis increments uniformly on a 6 hrs. time window.

In the hybrid variational assimilation system, the generated ensemble information has been projected into the **B** through the multivariate (temperature, salinity and sea surface height) vertical EOFs providing spatially varying daily estimates of $\mathbf{S}_e$, and $\mathbf{\Lambda}_e$. Ensemble information has been also used to compute daily varying horizontal correlation radii, $L_{x,y}^{\varepsilon}$, in $\mathbf{V}_H$. Several $\boldsymbol{\alpha}$ values have been tested. Sensitivity experiments have shown that the best results were obtained setting $\boldsymbol{\alpha}$=0.5, meaning that 50% of the vertical error covariance derives from the climatological statistics, while the remaining 50% derives from the ensemble statistics. In our hybrid system the observational representation error is proportional to the variance of the measures after binning in a 1 km square grid.

The ensemble statistics have been used also to estimate the model systematic error and a large scale systematic error correction has been applied. For every simulated day, $\bar{\mathbf{d}}$ has been computed using a depth depended observational window to avoid sampling error in the deep layers. The temporal window increases linearly from 11 days, at surface, to 25 days in the bottom layers. The resulting $\bar{\mathbf{d}}$ has been mapped onto the model grid ($\mathbf{P}^T$ in Eq. 17) by means of Barnes (Barnes, 1994) univariate objective analysis with smoothing length scales of 170 and 90km along x and y respectively. A length scale of 75 km has been used in the low pass filter ($L$ in Eq.17).

To test bias correction and the impact of the ensemble based EOFs, results from four different experiments are compared. Exp-ref uses climatological, spatially homogeneous **B** and no bias correction; Exp-Hy1 uses the hybrid, spatially and temporally varying, formulation of **B** but no bias correction; Exp-Cl1 uses the static formulation of **B** (as Exp-ref) but the bias correction is applied and finally Exp-Hy2 uses both the hybrid formulation of **B** and the bias correction. The differences between the experiments are summarized in Table 2.

## 4 Result and Discussion

The quality of the ensemble has been evaluated on the base of ensemble spread values and distributions. The ensemble spread is defined as the standard deviation across the ensemble members.

In Fig. 4, the time evolution of temperature and salinity standard deviations computed from the ensemble members are shown from the surface to 1000m depth. At all depths and for both temperature and salinity, the ensemble spread reaches a stable value on 10/12 June after 2/3 days assimilating the CTD and glider data from the first cruise leg (Fig.2). The small spread during the first days is mostly confined to the surface layers and is due to the SLA assimilation. Between 13 and 22 June the ensemble spread is nearly constant at all depths, probably constrained by the dense observational network, meaning that only a few days are needed to spin-up our ensemble system. Later on during the simulated period, the data density decreases and temperature and salinity ensemble spreads behaviors differ significantly. The salinity ensemble spread remains nearly constant while the standard deviation of surface temperature decreases. We can speculate that the decreased ensemble variability in temperature is due to the surface and lateral forcing shared among all the ensemble members that rapidly constrain the temperature within the model domain when no observations are assimilated. On the other hand, salinity reacts slower to surface forcing. Thus, the methodology used to generate the ensemble could be improved to also account for errors in the external forcing (surface and lateral open boundary conditions) and model parameterizations.

The horizontal distribution of the near surface (0.5 m depth) ensemble standard deviations for temperature and salinity valid for 12, 18, 24, and 30 June are shown in Fig. 5.

During the simulated period the two state variables show different behaviors. Temperature standard deviation maxima are mostly confined within the observational space and have well defined small/medium size structures. On 30 June, when no more *in-situ* observations are available, a large scale maxima structure is evident close to the north-west domain open boundaries partially due to SLA assimilation and the different structures and dynamics developed by the individual ensemble members approaching the open boundaries. On the other hand, salinity spread horizontal distributions are significantly different.

During the entire simulated period, maxima of the salinity ensemble spread are evident outside the area sampled by the observational campaign and structures are generally larger than in temperature. These are probably due to errors in the salinity content of waters masses forcing the simulations at the lateral open boundaries and conflicting with *in-situ* observations, thus generating fronts and instabilities. The adopted method to generate the ensemble members does not account for uncertainties in the forcing (surface or lateral) or initialization; further work is necessary in order to assess the impact of the forcing perturbation. The present work focuses on the potential benefits of a hybrid approach, rather than on evaluating the ensemble generation itself.

Fig. 6 shows an example of how the ensemble method changes the estimates of the salinity and temperature error vertical correlations and cross-correlations. On 22 June 2014, the ensemble estimates exhibit correlations of background temperature and salinity significantly different from the climatological estimate. Clearly, the ensemble method has added information to

the climatological estimates from the variability generated by the ensemble simulations on particular days. An interesting feature of temperature and salinity vertical error correlations on 22 June 2014 is the presence of several local maxima and minima. The similarities between static and ensemble-based correlations reflects the error in the large scale dynamical processes, introduced in our system by lateral open boundaries conditions. Salinity correlations (top right corner on both Fig.6 panels) show the largest differences between climatological/steady and the daily estimates. While the salinity climatological correlation field is characterized by generally positive values, in the daily estimate a clear anti-correlation pattern is observed starting at level 50 and persisting toward the bottom. This clearly indicates a more complex vertical error structure probably due to the presence of an intermediate water mass (the Modified Levantine Intermediate Waters) and deficiencies in the model to correctly simulate it. Similar patterns, even if less pronounced, are also observed in the temperature correlation and temperature-salinity cross-correlations. Furthermore, vertical scales of the correlations differ significantly. For instance, salinity vertical correlations are longer at the ocean bottom in case of the ensemble **B**, while an opposite feature is found for temperature. This suggests that the ensemble simulations lead to stronger consistency of the vertical cross-correlation at the ocean bottom between temperature and salinity with respect to the static **B**.

The temperature and salinity corrections due to the systematic error are shown in Fig.7. The panels on the left show how the vertical structure of the systematic error, averaged over the entire domain, evolve during the simulated period. In the right panels the maps of the systematic error correction averaged between 12 and 28 Jun at 100, 350 and 1000m depth are shown. During the first four days the number of *in-situ* observations increase and the spatial coverage improve. The systematic error computation and thus the corresponding correction is strongly affected by this observation sampling error. The sampling error is particularly evident in the surface and near surface corrections (between the surface and 300m depth), where scale of horizontal variability are small, that oscillate between positive and negative values. In the deeper layers the amplitude of this oscillation is significantly smaller. However, the overall effect of the correction after 4 days is to decrease the warm bias present in the deep temperature initial conditions and to increase salinity content at intermediate depths. At the end of the first cruise leg, 11 Jun, the systematic error stabilizes. After the initial shock due to the correction of the initial state, the systematic error correction corrects errors due to the surface forcing, the lateral open boundary condition and the inadequate model parameterizations.

The combined analysis of vertical structures and horizontal maps support some inferences. A thin layer with negative temperature correction is present between 5 and 15 m, the effect is to increase the stratification above 15m and decrease below. The model systematic error is clearly due to the vertical diffusion, however it is difficult to distinguish between error in the surface forcing or in the vertical turbulence closure scheme adopted. At 100 m depth the temperature correction is generally positive. The corrections map at this depth shows minimum values along the boundaries indicating that vertical mixing can be the source of the model failure. In the deeper layer the correction is more stable over the integrated time, and maps show maximum correction values close to the lateral open boundaries. Thus the adopted scheme is mostly acting on the external lateral forcing.

The surface salinity field reacts slowly to surface forcing. Simulation errors are mostly due to advection/diffusion processes. The salinity corrections in the first 100m are characterized by positive and negative values partially due to the observation sampling and systematic model error. At 100m depth the scheme increases the salinity content of the water masses along the southern open boundary with the exception of the southeast corner where negative corrections are found. This is probably due to a misplacements of the water masses present at this depth. At 300m depth the systematic error correction is generally positive and acts along the open boundaries. The simultaneous analysis of the temperature corrections indicates that warm and salty intermediate waters (the Modified Levantine Intermediate Waters) are poorly represented in the nesting model. At deeper layers the salinity correction is negative along the boundaries while it goes toward zero in the center of the model domain. We can argue that the vertical stratification of the nested model is too weak and tends to mix intermediate and deep water masses. The positive core in the center of the model domain suggests also that the choice of NEMO parameters for the vertical mixing is not optimal, leading to the vertical diffusion of the salt introduced with the data assimilation scheme.

The impacts of the daily ensemble based **B** and the bias correction on the quality of the simulations are evaluated comparing observations with model backgrounds.

In order to fully assess the performance of each experiment the Mean Squared Error (MSE) is decomposed following Oke et al. 2002 and the single components analyzed:

$$MSE = MB^2 + SDE^2 + 2S_m S_o (1 - CC)$$

where MB is the model mean bias, SDE the standard deviation error, $S_m$ and $S_o$ are the modelled and observed standard deviations and CC is the cross-correlation between modelled and observed fields. The skill of each experiment with respect to a reference experiment (Exp-ref), is calculated based on the MSE. The skill score (SS; e.g. Murphy 1989) is defined as:

$$SS = 1 - \frac{MSE}{MSE_r}$$

Where $MSE_r$ is the MSE of Exp-ref. The normalized (using the observed standard deviation) root mean square error, mean bias and standard deviation error together with the cross-correlation and Skill Score vertical profiles for temperature and salinity and for the four experiments are shown in Fig.8 and Fig.9 respectively. In table 3 the non-normalized statistics vertically integrated at predefined layers are also listed. All the statistics are computed using data and corresponding model backgrounds collected during the second cruise leg started on 12 June and ended 25 June 2014

The analysis of the single components of the model error allows us to identify the effect of the bias correction procedure and the impact of the daily, ensemble based, estimate of vertical covariance. The two simulations without the bias correction (Exp-ref and Exp-Hy1) are characterized by a similar vertical structure and values of temperature and salinity RMSE (Fig.8 and Fig.9 panels A for temperature and salinity respectively). Both the simulations are characterized by a large temperature RMSE

below 500m depth, while they both show a maximum in salinity RMSE at about 400m. Large errors are also observed in temperature between 60 and 200m depth and in salinity between 40 and 150m depth. The vertical profiles of the mean bias (panels B of Fig.8 and Fig.9) clearly indicates that at intermediate and deep layers this error structure is due to a large bias characterizing the system (Initial state and lateral open boundary conditions). The near surface RMSE maxima do not have a
clear correspondence in the mean bias structure. The bias nature of this error is confirmed by the results obtained with the two experiments where the bias correction has been applied. This also confirms that our simple systematic error correction procedure is capable of significantly reducing this bias. Both Exp-Hy2 and Exp-Cl2 RMSE and mean bias are characterized by a nearly uniform and relatively small values. However, the systematic error correction increases the temperature mean bias between 20 and 70m depth, meaning that scales (both spatial and temporal), procedure or observation sampling used are
probably not adequate at these depths. On the other hand, at similar depths, the systematic error correction reduces the salinity mean bias (Fig.9 B). We argue that temperature and salinity systematic errors in these layers have different length scales.

The standard deviation error indicates the capability of our system to correctly reproduce the amplitude of the observed spatial/temporal variability. Differences between climatological and daily estimate of the background error covariance are evident. The usage of daily hybrid **B** without the bias correction introduces in the system a large temperature standard deviation
error between 60 and 150m depth, significantly larger than in Exp-Ref. It is interesting to note that the same vertical error statistic (**B**) when applied together with the bias correction procedure (Exp-Hy2) reduces significantly the standard deviation error at the same depths.

The differences introduced by the daily, ensemble based, estimates of the background vertical error covariance are evident analyzing the cross-correlation (panels D of Fig.8 and Fig.9 for temperature and salinity respectively) and the skill scores
(panels E). The Exp-Hy2 with systematic error correction and daily estimate of the vertical error background covariance has a temperature cross-correlation generally higher than the other experiments. These differences are maximum between 20 and 80m and below 250m depth. On the other hand in the salinity field the maximum differences are observed near the surface (between 0 and 50m depth) while in the deeper layers Exp-Cl1 and Epx-Hy2 perform in a similar way. Both the experiments with the bias correction show a decreased cross-correlation with observed salinity between 200 and 400m depth.
The overall experiment statistics are listed in Table 3. Exp-Hy2 vertically integrated temperature skill score is 55%; 47% is due to the systematic error correction (Exp-Cl1 SS is .47) while the remaining part is due to the introduction of the daily ensemble based estimates of **B**. The simple introduction of the ensemble based **B** (Exp-Hy1) produces a worsening of the solution between 50 and 215m depth. We argue that the small structures introduced with the assimilation scheme are not in balance with the surrounding environment and develop incorrect dynamics; the correction of the systematic error allow the
model to incorporate the information provided.

The vertically integrated salinity Exp-Hy2 and Exp-Cl1 skill scores are 42% however they have different vertical distributions. Exp-Hy2 strongly outperforms Exp-Cl1 in the surface layers (0-50m depth), while it does not improve significantly the model solution between 110 and 215 m depth. In the other layers the two systems have similar performance. The improvements and

the worsening are both due to the cross-correlation between observations and modelled salinities. This can be a consequence of the relatively small ensemble size that has not adequately sampled the model error.

## 5 Summary and Conclusions

During June 2014 an extensive sea-trial (Recognized Environmental Picture, REP14-MED) off the west coast of Sardinia was conducted by CMRE (Centre for Maritime Research and Experimentation). Two research vessels and a glider fleet collected a massive amount of data in an area of approximatively 10000 km$^2$. Remote sensing data and existing products from atmospheric and oceanographic operational models were also collected as an additional observational dataset and boundary conditions, respectively.

A Nucleus of European Modeling of the Ocean (NEMO, Madec et al. 2008) based model has been implemented in the area with a horizontal resolution of approximatively 1km and 91 hybrid vertical levels. The model has been initialized and forced at the lateral open boundaries using Mercator-Ocean (Drévillon et al. 2008) daily analyses, while atmospheric forcing was computed by means of interactive bulk formulae (Oddo et al. 2009) using the hourly operational products from the COSMO-ME limited area atmospheric model.

In order to address the data assimilation issues characterizing ocean limited area models with dense observational networks a 3DVar assimilation scheme was implemented and coupled with the NEMO based code. Following Dobricic and Pinardi (2008) the present variational scheme decomposes the Background Error Covariance matrix (**B**) in a sequence of linear operators each of them representing a specific component of the error structure. Two main issues have been encountered in the present assimilation exercise. The first is related to the small scales sampled by the dense observation network which are poorly represented in the traditionally stationary vertical component of **B**. The second concerns the large systematic errors partially introduced by the external forcing (experiment initialization, lateral or surface open boundary conditions) and partially due to inadequate model physics. In order to overcome these limits and improve the system, a variational-ensemble hybrid assimilation system has been developed and implemented. A small size ensemble (14 members) has been created by combining perturbation of observations and background-error horizontal correlation radii in the **B** matrix. The choice of creating the ensemble members by perturbing only the analyses is mostly justified by the nature of the experiment we conducted. In fact a perturbation in the model initialization or model parameterizations would require a relatively long integration time in order to fully develop and reach a stable condition. On the other hand, perturbing the observations on a daily assimilation system allows us to quickly generate ensemble statistics with amplitudes similar to the model error. The statistical information retrieved from the ensemble members has been used to address both the small scale and the systematic error issues. In order to improve the representation of the small scale error in the background-error covariances, the climatological based $V_v$ operator (accounting for the multivariate vertical background error covariances) has been replaced with a daily and spatially varying estimate computed by applying multivariate EOFs analysis to the ensemble members' anomalies. Furthermore, the climatological

estimates of the recursive filter horizontal correlation radii used to model the $\mathbf{V_H}$ operator have been substituted with daily estimates computed from the ensemble statistics according Belo Pereira and Berre (2006).

To correct the systematic error the ensemble members' misfit statistics have been used. For every simulated day an estimate of the systematic error has been obtained by averaging the misfit over the ensemble members and assuming that observational error and random model error have both zero mean. The results have been mapped onto the model grid using a univariate objective analysis (Barnes 1994) and superimposed to the ensemble daily mean. At each assimilation step the differences between the corrected ensemble mean and the last available daily average corresponding fields have been filtered with a low pass filter with 75 km length scale and the results superimposed to the 3Dvar corrections.

The implementation of our strategy suffers from the need to empirically choose the parameters associated with the combination of stationary and ensemble-derived covariances and with the scales for the large-scale bias estimation, which can both benefit from further tuning in the future. However, these experiments represent a proof-of-concept for including flow-dependent and large-scale aspects in a variational assimilation framework.

In order to test the validity of our hypothesis and to quantitatively estimate the differences introduced with the hybrid-variational scheme designed, the results of 4 different experiments have been compared. Exp-ref uses the standard 3DVar scheme with static and homogeneous $\mathbf{V_v}$ and $\mathbf{V_H}$ both computed using 15 years Mediterranean CMEMS reanalysis (Adani et al. 2011). In Exp-Hy1 the climatological $\mathbf{V_v}$ and $\mathbf{V_H}$ have been weighted with daily estimates from the ensemble statistic, with $\mathbf{V_v}$ also varying spatially. Exp-Cl1 uses the same $\mathbf{B}$ formulation of Exp-ref but the systematic error correction procedure has been applied. Finally Exp-Hy2 uses the same $\mathbf{B}$ formulation of Exp-Hy1 but the systematic error correction procedure has also been applied. The simple introduction of the hybrid estimate of $\mathbf{B}$ does not significantly improve the model results. This is probably due to the relatively small ensemble size and the amplitude of the large scale systematic error characterizing our experiments. The vertically integrated Skill Score of Exp-Hy1 with respect to Exp-ref is 0.06 for temperature and 0 for salinity, indicating improvements in temperature Mean Error of 6% and no improvements in the salinity field. However a significant worsening of the model temperature is observed between 50 and 100m. The systematic error correction accounts for a large part of the improvements and the ensemble based estimates of $\mathbf{B}$ produce the best results when used in combination with the systematic error correction (Exp-Hy2). We can argue that the small scale corrections introduced with the new formulation of $\mathbf{B}$ are not in balance with the surrounding environment and thus not properly ingested into the model solution, thus requiring the additional large-scale bias correction. In both Exp-Cl1 and Exp-Hy2 the systematic error correction correctly reduces the large warm bias affecting the temperature initial state and lateral open boundary condition below 500m and simultaneously removes the salinity error at intermediate depth due to the absence in the external data of the correct water masses at this depths.

The adopted methodology seems to produce satisfactory results. During the first days, with the observational data availability increasing, the systematic error oscillates and finally adjusts the errors associated to the initial conditions of the experiments. The amplitude of the corrections during this initial phase is relatively large. After the errors due to the initialization have been reduced the amplitude of the systematic error correction significantly reduces and acts mostly on the lateral open boundary conditions. The improvements in the Exp-Hy2 are mostly due to improvements in the Cross-Correlation and thus to a better

reproduction of horizontal and vertical dynamics and structures. In terms of SS for temperature Exp-Hy2 performs best at all the depths, with an overall improvements of 55% with respect to Exp-Ref while Exp-Cl1 vertically integrated temperature improvement is 47%. Large parts of differences between Exp-Hy2 and Exp-Cl1 can be traced back to an improved cross-correlation coefficient between modelled and observed values. The salinity statistics show different model behaviors. The vertically integrated SS are similar for Exp-Cl1 and Exp-Hy2, both improve Exp-Ref results by about 42%. However the distribution of the error differs significantly along the vertical. Exp-Hy2 outperform Exp-Cl2 in the first 100m of the water column as consequence of larger cross-correlation coefficient with observations, while Exp-Cl2 perform better in the intermediate layers (between 110 and 215m). It should be noted that the idea of using the current information from misfits or from the ensemble to improve the analysis has been recently applied in several other studies (Wang *et al.* 2007 and 2008, Desroziers *et al.* 2006, Hamill and Snyder 2000, Etherton and Bishop 2004). However, in all the methods, and also in schemes presented and adopted in this manuscript the weights given to the climatological and ensemble based **B** estimates are arbitrary. The vertical dependency of the hybrid system's performance suggests that the empirical methods used in the estimates of the ensemble size and the relative weights of static and hybrid **B** require a more objective and formal approach. The two quantities are clearly correlated. In this study we used a small ensemble size (14 members) and constant (both spatially and temporally) weights obtaining, however, encouraging results. Recently Dobricic et al. (2015) overcome this issue by proposing a method based on Bayesian Hierarchical Model where the relative weights arise directly from the computations based on Bayes' theorem.

There are several possible future improvements of the hybrid variational scheme method presented for estimating background-error covariances. Menetrier and Auligne' (2015) suggest a theoretical framework where hybrid weights and parameters for the localization of ensemble derived covariances are jointly optimized as a function of the ensemble size. An alternative possibility may be to include the $\alpha$ parameter in the minimization of the cost function obtaining an optimized and variable relative weight.

**Acknowledgements**

CMRE activities have been supported by the NATO Allied Command Transformation (ACT) through the contract SAC000404. The Authors wish to thank Masters and crews of NR/V Alliance (CMRE) and R/V Planet (German Ministry of Defence), as well as the many colleagues, both from CMRE and the Wehrtechnische Dienststelle für Schiffe und Marinewaffen, Maritime Technologie und Forschung (WTD71; Germany), which contributed to provide the assimilated CTD and glider (3 from CMRE and 1 from WTD71) data. In particular the Authors wish to thank Dr. Heinz-Volker Fiekas, scientist in charge on-board the R/V Planet, and Dr. Michaela Knoll, who managed the R/V Planet CTD dataset.

*APPENDIX*

We start from the cost function:

$$J(\delta x) = \frac{1}{2}\delta x^{\mathrm{T}}\mathbf{B}^{-1}\delta x + \frac{1}{2}(\mathbf{H}\delta x - \mathbf{d})^{\mathrm{T}}\mathbf{R}^{-1}(\mathbf{H}\delta x - \mathbf{d}) \tag{A.1}$$

To define our hybrid assimilation schemes we compute $\mathbf{B}$ as a linear combination of the "static" covariance operator, $\mathbf{B}_c$, and the flow-dependent operator, $\mathbf{B}_e$:

$$\mathbf{B} = \alpha\mathbf{B}_c + (1 - \alpha)\mathbf{B}_e \tag{A.2}$$

where $\alpha$ is the relative weight. Substituting A.2 in A.1 we obtain the new hybrid cost function:

10  $$J(\delta x) = \frac{1}{2}\delta x^{\mathrm{T}}(\alpha\mathbf{B}_c + (1 - \alpha)\mathbf{B}_e)^{-1}\delta x + \frac{1}{2}(\mathbf{H}\delta x - \mathbf{d})^{\mathrm{T}}\mathbf{R}^{-1}(\mathbf{H}\delta x - \mathbf{d}) \tag{A.3}$$

We define now the increment as a weighted sum of parts corresponding to static and flow-dependent covariance matrices:

$$\delta x = \delta x_c + \delta x_e.$$

We want to demonstrate that:

$$J(\delta x) = \frac{1}{2}\delta x_c^{\mathrm{T}}(\alpha\mathbf{B}_c)^{-1}\delta x_c + \frac{1}{2}\delta x_e^{\mathrm{T}}((1 - \alpha)\mathbf{B}_e)^{-1}\delta x_e + \frac{1}{2}(\mathbf{H}\delta x - \mathbf{d})^{\mathrm{T}}\mathbf{R}^{-1}(\mathbf{H}\delta x - \mathbf{d}) \tag{A.4}$$

has the minimum for the same value of $\delta\mathbf{x}$ as A.3.

To minimize A.4, $\delta x_c$ and $\delta x_e$ must satisfy $\frac{\partial J(\delta x)}{\partial x_c} = 0$ and $\frac{\partial J(\delta x)}{\partial x_e} = 0$ which gives:

$$(\alpha\mathbf{B}_c)^{-1}\delta x_c + \frac{\partial}{\partial x_c}\left(\frac{1}{2}\delta x_e^{\mathrm{T}}((1 - \alpha)\mathbf{B}_e)^{-1}\delta x_e\right) + \frac{1}{2}\frac{\partial J_o}{\partial x_c} = 0 \tag{A.5}$$

$$[(1 - \alpha)\mathbf{B}_e]^{-1}\delta x_e + \frac{\partial}{\partial x_e}\left(\frac{1}{2}\delta x_c^{\mathrm{T}}(\alpha\mathbf{B}_c)^{-1}\delta x_c\right) + \frac{1}{2}\frac{\partial J_o}{\partial x_e} = 0 \tag{A.6}$$

where Jo is the observational term. Assuming that $\delta x_c$ and $\delta x_e$ can be perturbed independently, both the second terms on the left hand side of A.5 and A.6 are null:

$$\frac{\partial}{\partial x_c}\left(\frac{1}{2}\delta x_e^{\mathrm{T}}((1 - \alpha)\mathbf{B}_e)^{-1}\delta x_e\right) = 0, \tag{A.7}$$

$$\frac{\partial}{\partial x_e}\left(\frac{1}{2}\delta\boldsymbol{x_c}^T(\alpha\mathbf{B_c})^{-1}\delta\boldsymbol{x_c}\right) = 0 \tag{A.8}$$

and

5    $$\frac{\partial J_o}{\partial x} = \frac{\partial J_o}{\partial x_e} = \frac{\partial J_o}{\partial x_c} = 2\mathbf{H}^T\mathbf{R}^{-1}(\mathbf{H}\delta\boldsymbol{x} - \mathbf{d}) . \tag{A.9}$$

This is a reasonable assumption, because the two random values are sampled from different Gaussians. Although they are defined over the same space, one is sampled from historical states, and the other from current forecasts. Premultiplying A.5 by $\alpha\mathbf{B_c}$ and A.6 by $(1-\alpha)\mathbf{B_e}$, removing the null terms, summing the two subsequent equations and applying A.9 yields:

$$0 = (\delta\boldsymbol{x_c}+\delta\boldsymbol{x_e}) + \frac{1}{2}[\alpha\mathbf{B_c} + (1-\alpha)\mathbf{B_e}]\,\frac{\partial J_o}{\partial x} \tag{A.10}$$

Multiplying A.10 by the inverse of the hybrid covariance:

$$0 = [\alpha\mathbf{B_c} + (1-\alpha)\mathbf{B_e}]^{-1}(\delta\boldsymbol{x_c}+\delta\boldsymbol{x_e}) + \mathbf{H}^T\mathbf{R}^{-1}[\mathbf{H}(\delta\boldsymbol{x_c}+\delta\boldsymbol{x_e}) - \mathbf{d}] \tag{A.11}$$

This is also the minimum of A.3 that we wanted as a proof.

Furthermore, defining the background and analysis perturbations around the true state $\boldsymbol{x_t}$ as:

$$\delta\boldsymbol{x_b} = \boldsymbol{x_b} - \boldsymbol{x_t}$$

and

$$\delta\boldsymbol{x_a} = \boldsymbol{x_a} - \boldsymbol{x_t},$$

by adding and subtracting the true state A.11 becomes*:*

$$0 = [\alpha\mathbf{B_c} + (1-\alpha)\mathbf{B_e}]^{-1}(\boldsymbol{x_a} - \boldsymbol{x_b} - \boldsymbol{x_t} + \boldsymbol{x_t}) + \mathbf{H}^T\mathbf{R}^{-1}[\mathbf{H}(\boldsymbol{x_a} - \boldsymbol{x_b} - \boldsymbol{x_t} + \boldsymbol{x_t}) - (\boldsymbol{y} - \boldsymbol{Hx_b})]$$

or:

$$0 = [\alpha\mathbf{B_c} + (1-\alpha)\mathbf{B_e}]^{-1}(\delta\boldsymbol{x_a} - \delta\boldsymbol{x_b}) + \mathbf{H}^T\mathbf{R}^{-1}[\mathbf{H}\delta\boldsymbol{x_a} - (\boldsymbol{y} - \boldsymbol{Hx_t})]$$

that can be written also as:

$$\{[\alpha\mathbf{B}_c + (1-\alpha)\mathbf{B}_e)]^{-1} + \mathbf{H}^T\mathbf{R}^{-1}\mathbf{H}\}\delta x_a = [\alpha\mathbf{B}_c + (1-\alpha)\mathbf{B}_e)]^{-1}\delta x_b + \mathbf{H}^T\mathbf{R}^{-1}[y - \mathbf{H}x_t]. \qquad \text{A.12}$$

Multiplying each side of A.12 by its transpose, taking the expectation, assuming that observational errors are independent of background errors:

$$\{[\alpha\mathbf{B}_c + (1-\alpha)\mathbf{B}_e)]^{-1} + \mathbf{H}^T\mathbf{R}^{-1}\mathbf{H}\}A\{[\alpha\mathbf{B}_c + (1-\alpha)\mathbf{B}_e)]^{-1} + \mathbf{H}^T\mathbf{R}^{-1}\mathbf{H}\}^T = [\alpha\mathbf{B}_c + (1-\alpha)\mathbf{B}_e)]^{-1}E\{\delta x_b(\delta x_b)^T\}[\alpha\mathbf{B}_c + (1-\alpha)\mathbf{B}_e)]^{-T} + \mathbf{H}^T\mathbf{R}^{-1}\mathbf{R}\mathbf{R}^{-1}\mathbf{H}. \qquad \text{A.13}$$

Assuming the **B** contains the true background error covariances, i.e. the background errors are well specified, and using A.2:

$$E\{\delta x_b(\delta x_b)^T\} = \alpha\mathbf{B}_c + (1-\alpha)\mathbf{B}_e$$

thus:

$$\{[\alpha\mathbf{B}_c + (1-\alpha)\mathbf{B}_e)]^{-1} + \mathbf{H}^T\mathbf{R}^{-1}\mathbf{H}\}A\{[\alpha\mathbf{B}_c + (1-\alpha)\mathbf{B}_e)]^{-1} + \mathbf{H}^T\mathbf{R}^{-1}\mathbf{H}\}^T = [\alpha\mathbf{B}_c + (1-\alpha)\mathbf{B}_e)]^{-1}[\alpha\mathbf{B}_c + (1-\alpha)\mathbf{B}_e)][\alpha\mathbf{B}_c + (1-\alpha)\mathbf{B}_e)]^{-T} + \mathbf{H}^T\mathbf{R}^{-1}\mathbf{R}\mathbf{R}^{-1}\mathbf{H}. \qquad \text{A.14}$$

or :

$$\{[\alpha\mathbf{B}_c + (1-\alpha)\mathbf{B}_e)]^{-1} + \mathbf{H}^T\mathbf{R}^{-1}\mathbf{H}\}A\{[\alpha\mathbf{B}_c + (1-\alpha)\mathbf{B}_e)]^{-1} + \mathbf{H}^T\mathbf{R}^{-1}\mathbf{H}\}^T = [\alpha\mathbf{B}_c + (1-\alpha)\mathbf{B}_e)]^{-1} + \mathbf{H}^T\mathbf{R}^{-1}\mathbf{H}.$$

Dividing by $\{[\alpha\mathbf{B}_c + (1-\alpha)\mathbf{B}_e)]^{-1} + \mathbf{H}^T\mathbf{R}^{-1}\mathbf{H}\}$:

$$A = \{[\alpha\mathbf{B}_c + (1-\alpha)\mathbf{B}_e)]^{-1} + \mathbf{H}^T\mathbf{R}^{-1}\mathbf{H}\}^{-1} \qquad \text{A.15}$$

where $A = E\{\delta x_a(\delta x_a)^T\}$ is the analysis error covariance matrix. A.15 demonstrates that independent forecasts updates in each ensemble member by using A.4 give the same optimal estimate of updated covariances as A.3.

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

*TABLES:*

| | |
|---|---|
| **N. points** *x/y/z* | 235/246/91 |
| *Δx / Δy* | ~1000 m |
| *ΔT* **time-step** | 100 s |
| **Vertical Discretization** | Hybrid z-sigma (Siddorn and Furner 2012) |
| **Surface Fluxes** | MFS Bulk (Oddo et al 2009) |
| **Atmospheric data** | Hourly COSMO |
| **SST** | GOS_SST -30 W/m2/K |
| **Lateral boundary condition (LBC)** | No slip |
| **Open LBC (barotropic/baroclinic/tracers)** | Flather (1976) / imposition / Neumann |
| **Bottom friction** | Non-linear |
| **EOS** | EOS-80 |
| **Tracer Advection** | TVD (Zalesak, 1979) |
| **Tracer Diffusion** | Laplacian along iso-surface $K_h$=10 |
| **Momentum Advection** | Vector form |
| **Momentum Diffusion** | Bilaplacian along iso-surface $A_h$= -2.5 $e^7$ |
| **Vertical turbulence** | GLS with Canuto A (2001) and k-eps |
| **Free surface** | Filtered (Madec et al 2008) |

**Table 1** *Model Configuration details*

|  | $V_v$ | Bias corr. | H. Corr Radii |
|---|---|---|---|
| *Exp-ref* | **Static-Clim** | **No** | **Const** |
| *Exp-Hy1* | **Hybrid** | **No** | **Variable** |
| *Exp-Cl1* | **Static-Clim** | **Yes** | **Const** |
| *Exp-Hy2* | **Hybrid** | **Yes** | **Variable** |

**Table 2** *Experiments*

| layer | ref | Hy1 | Cl1 | Hy2 | ref | Hy1 | Cl1 | Hy2 | ref | Hy1 | Cl1 | Hy2 | ref | Hy1 | Cl1 | Hy2 |
|---|---|---|---|---|---|---|---|---|---|---|---|---|---|---|---|---|
| | MB | | | | SDE | | | | CC | | | | SS | | | |
| *Temperature* | | | | | | | | | | | | | | | | |
| 0-50 | -.05 | **0.02** | 0.11 | 0.16 | **0.02** | **-.02** | **0.02** | -.06 | -.01 | 0.15 | 0.17 | **0.22** | / | 0.19 | 0.07 | **0.21** |
| 50-110 | 0.10 | 0.13 | **0.03** | **0.03** | 0.10 | 0.16 | 0.05 | **0.03** | 0.19 | **0.26** | 0.17 | 0.19 | / | -.50 | 0.37 | **0.47** |
| 110-215 | 0.09 | 0.08 | -.05 | **-.03** | **-.04** | **-.04** | -.06 | -.08 | 0.24 | 0.17 | **0.26** | 0.25 | / | -.06 | 0.20 | **0.29** |
| 215-470 | 0.06 | **0.04** | -.06 | **-.04** | **-.12** | **-.12** | **-.12** | -.14 | 0.07 | 0.18 | 0.21 | **0.34** | / | 0.11 | 0.10 | **0.25** |
| 470-930 | 0.41 | 0.40 | -.03 | **-.02** | -.10 | -.10 | **-.02** | -.04 | 0.04 | 0.37 | 0.46 | **0.51** | / | 0.11 | 0.81 | **0.84** |
| 0-930 | 0.23 | 0.22 | -.03 | **-.01** | -.08 | -.08 | **-.05** | -.07 | 0.08 | 0.27 | 0.33 | **0.39** | / | 0.06 | 0.47 | **0.55** |
| *Salinity* | | | | | | | | | | | | | | | | |
| 0-50 | **-.01** | -.02 | **0.01** | **0.01** | **-.01** | **-.01** | -.02 | -.02 | 0.23 | 0.07 | 0.36 | **0.61** | / | -0.24 | 0.28 | **0.52** |
| 50-110 | -.01 | -.01 | 0.01 | **0.00** | 0.02 | 0.02 | **0.00** | **0.00** | 0.14 | 0.23 | **0.64** | 0.62 | / | 0.10 | 0.64 | **0.66** |
| 110-215 | 0.02 | **0.00** | -.01 | -.02 | **-.02** | **-.02** | **-.02** | **-.02** | 0.14 | -0.01 | **0.19** | 0.09 | / | -0.04 | **0.08** | 0.01 |
| 215-470 | -.10 | -.10 | **-.02** | **-.02** | -.04 | -.04 | **-.03** | -.04 | 0.08 | **0.20** | 0.09 | 0.07 | / | -0.02 | **0.62** | **0.62** |
| 470-930 | -.05 | -.05 | **-.01** | **-.01** | -.02 | -.02 | **-.01** | **-.01** | 0.44 | 0.49 | **0.52** | 0.46 | / | 0.03 | **0.38** | 0.36 |
| 0-930 | -.05 | -.05 | **-.01** | **-.01** | **-.02** | **-.02** | **-.02** | **-.02** | 0.27 | 0.31 | **0.36** | 0.33 | / | 0.00 | **0.42** | **0.42** |

**Table 3 Model Bias (MB), Standard Deviation Error (SDE), Cross-Correlation (CC) and Skill Score (SS) for the different experiments and integrated between different layers. For each quantity the best performing models is highlighted in bold.**

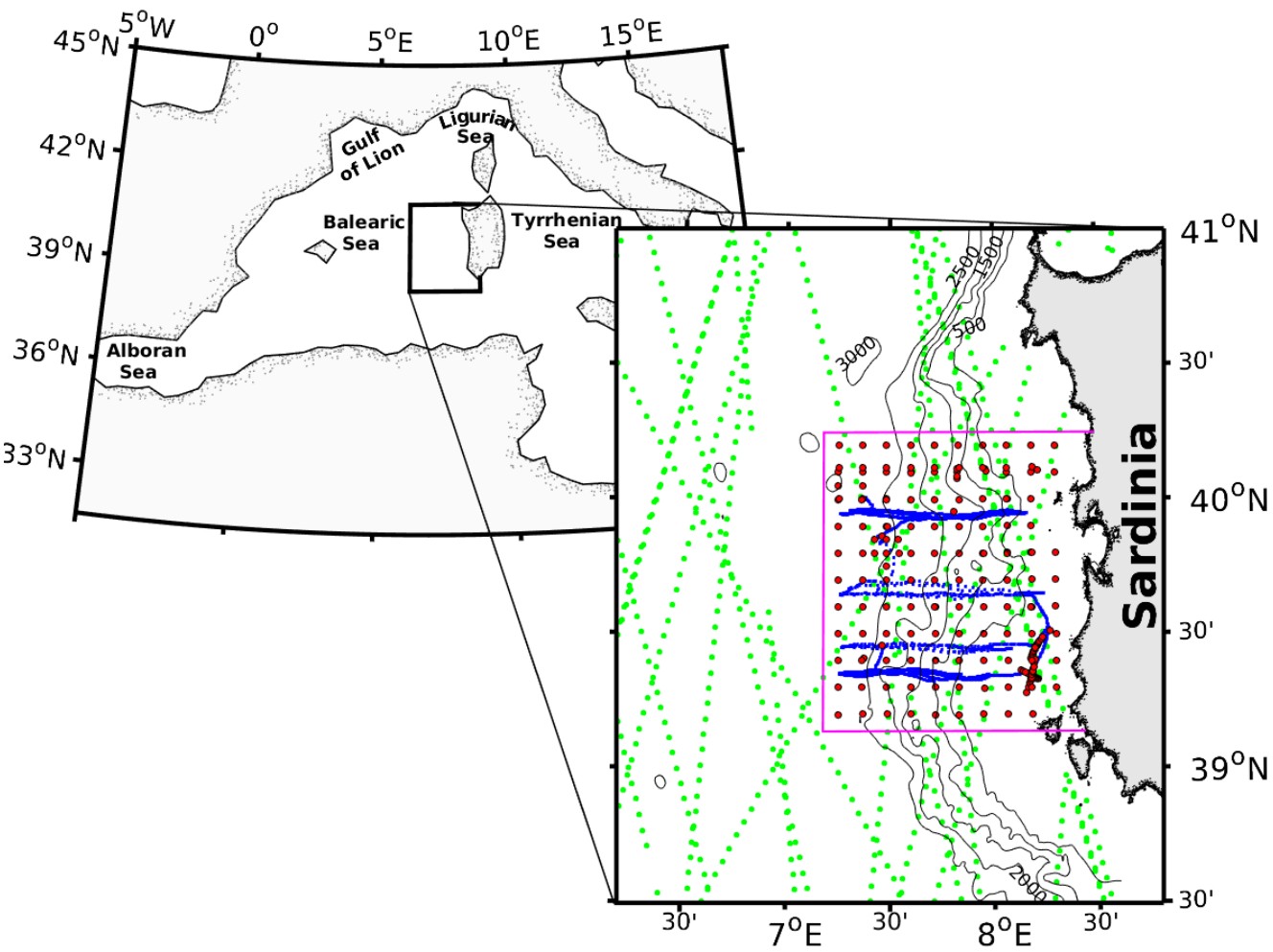

**Figure 1: Top-Left panel: Western Mediterranean Sea. Bottom-right panel: Model domain and collected data during REP14-MED experiment. Green dots indicate Sea Level Anomaly measurements from satellite (from 3 to 30 Jun 2014), red dots CTD positions (from 7 to 24 Jun 2014), blue dots glider trajectories (surfacing points, from 8 to 23 Jun 2014). The magenta lines indicate the box used to compute ensemble statistics. Bathymetric lines are also shown (m).**

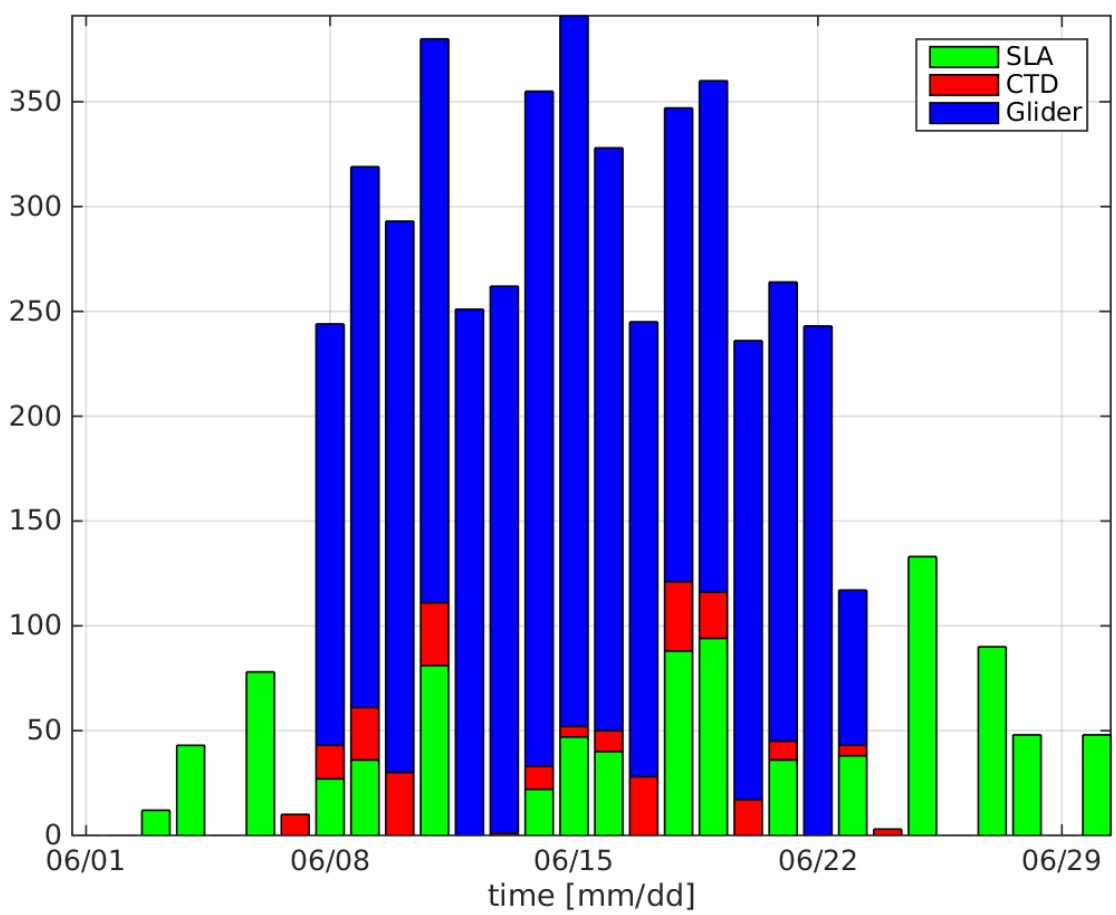

**Figure 2: Number of observations per day. The color coding is according to Fig. 1. The y-axis indicates the number of vertical profiles for CTD and gliders and number of points within the model domain for the SLA data. The x-axis indicates time.**

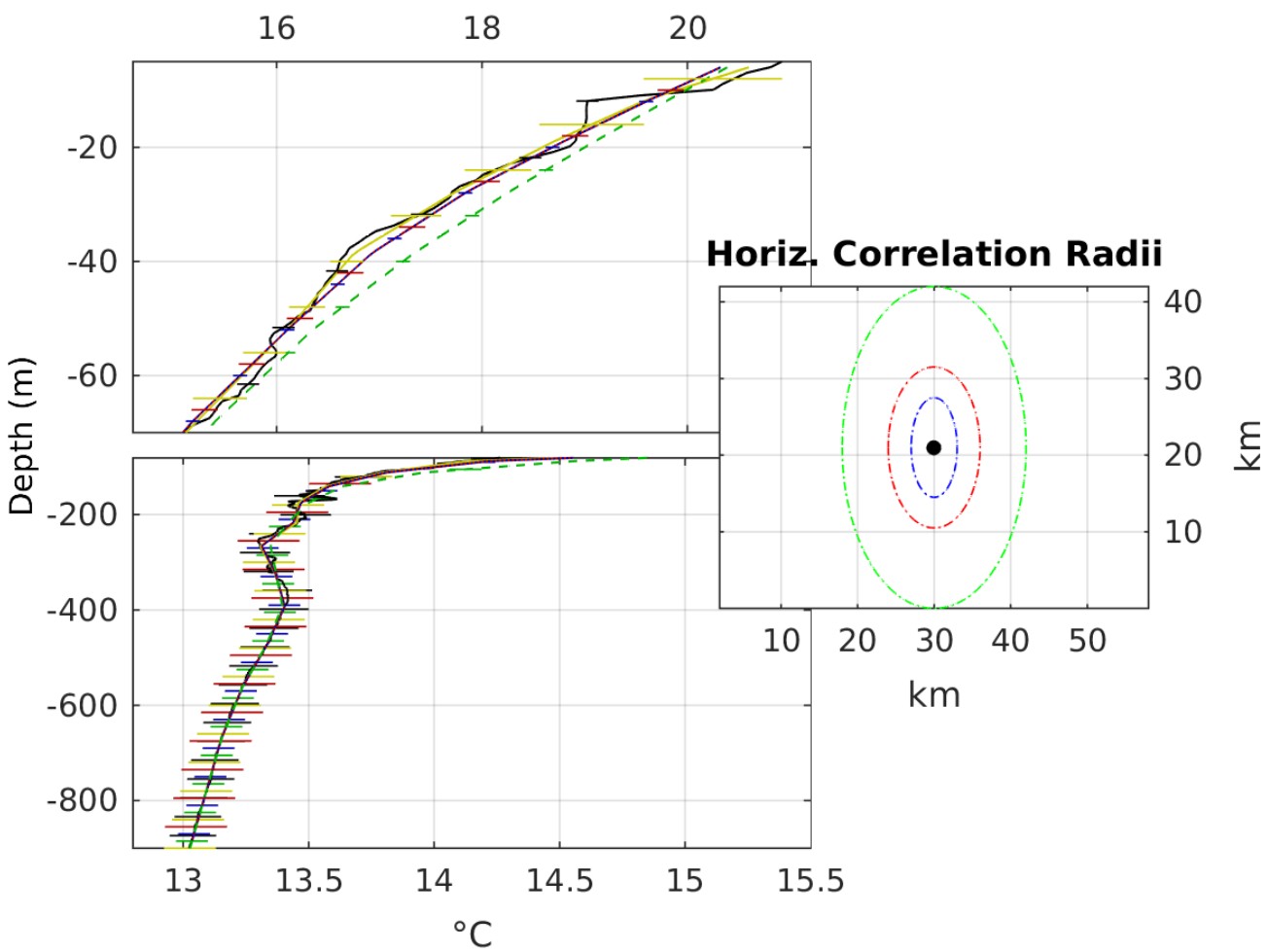

**Figure 3: Example of perturbed CTD vertical profile with different quality check procedure and filtering applied. The solid black line indicates the full resolution CTD profile while horizontal lines are the associated observational error. The other colours indicate the perturbed profile. In the middle panel the 3 tested couples of horizontal correlation length scales ($L^{\varepsilon}_{(x,y)}$) are shown. The circles indicate the distance where the horizontal correlation of a single observation is zero. The green circle length scales are $L^{\varepsilon}_{x,}$ =12 and $L^{\varepsilon}_{y}$ =21 km, this set has been used also in the reference experiment. The red circle radii are $L^{\varepsilon}_{x,}$=6 and $L^{\varepsilon}_{y}$=12 km. The blues circle radii are $L^{\varepsilon}_{x,}$=3 and $L^{\varepsilon}_{y}$=6 km.**

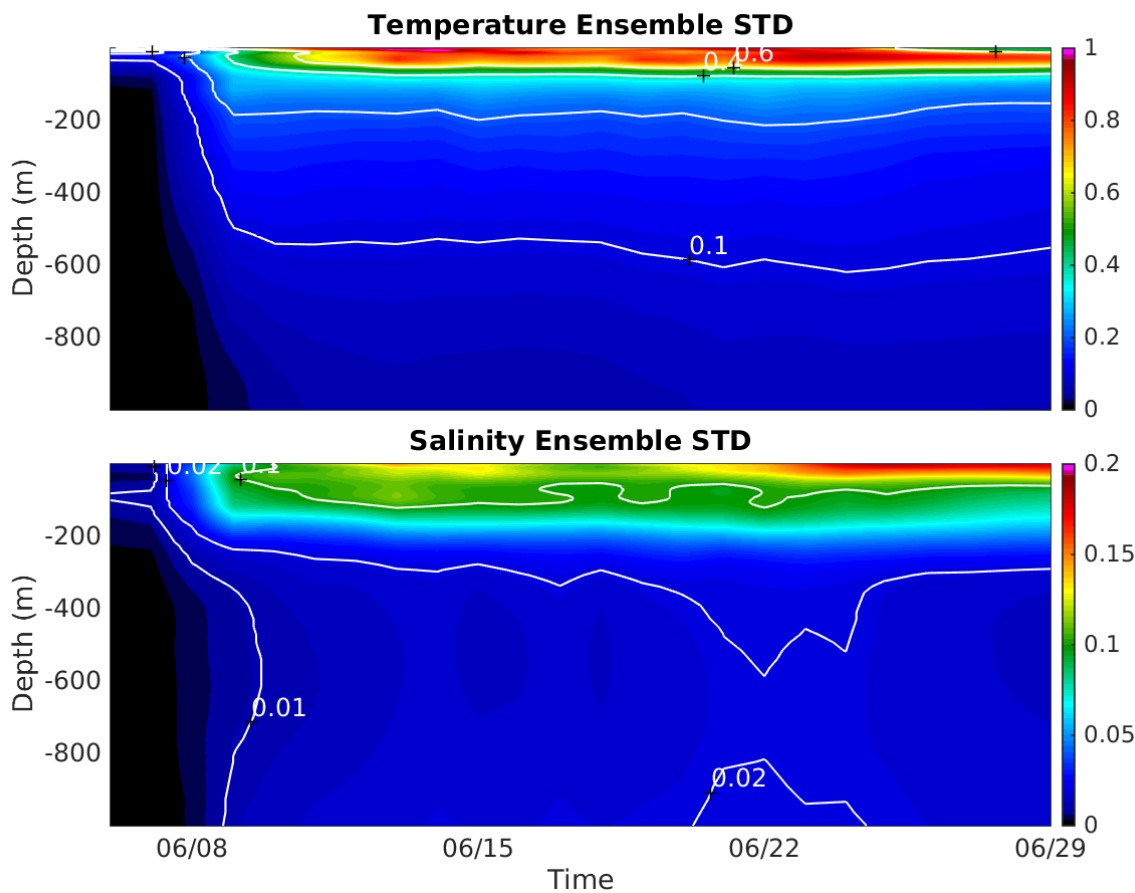

**Figure 4: Ensemble Standard deviation computed over the observational domain in the observational box (magenta) shown in Fig.1. Ordinate indicates ocean depth in meter while abscissa is time.**

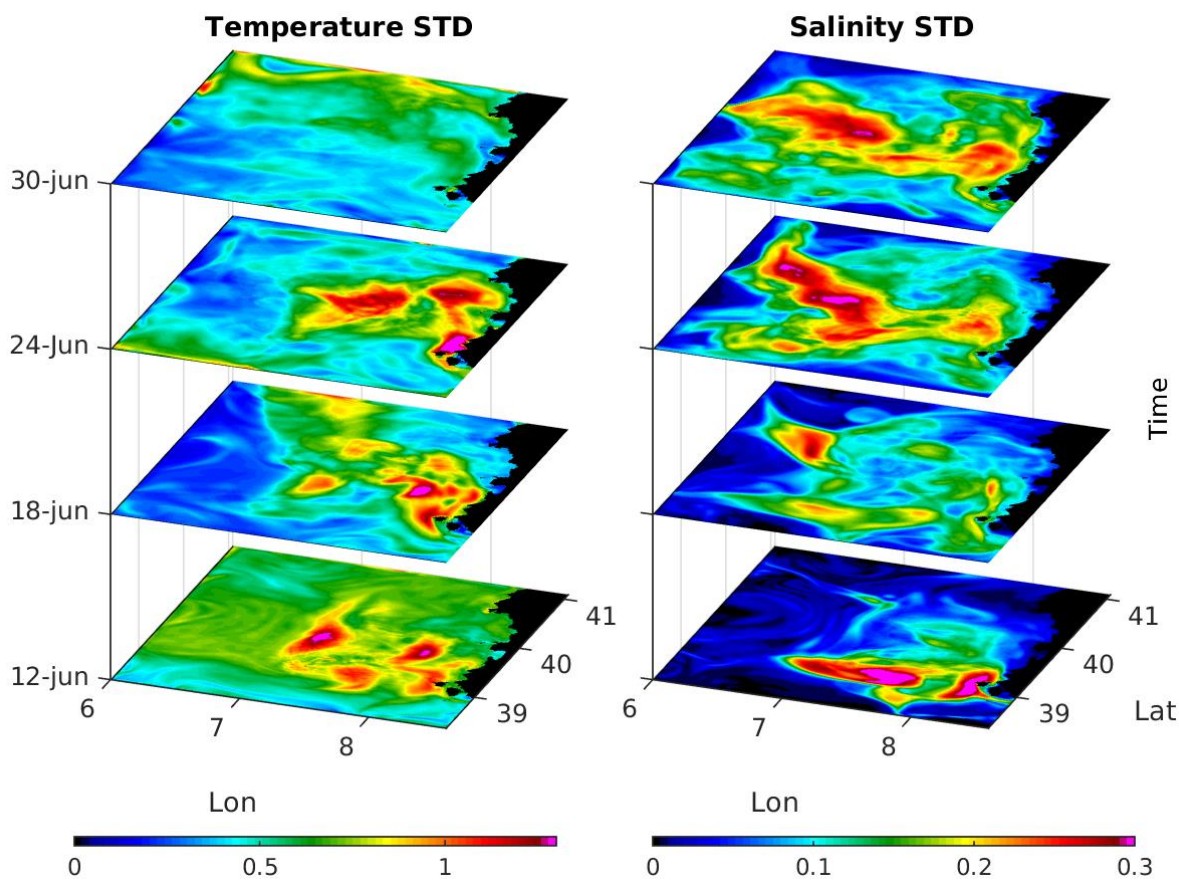

**Figure 5: Horizontal maps of ensemble standard deviation for temperature (left) and salinity (right) at 0.5-m on 12, 18, 24 and 30 June 2014.**

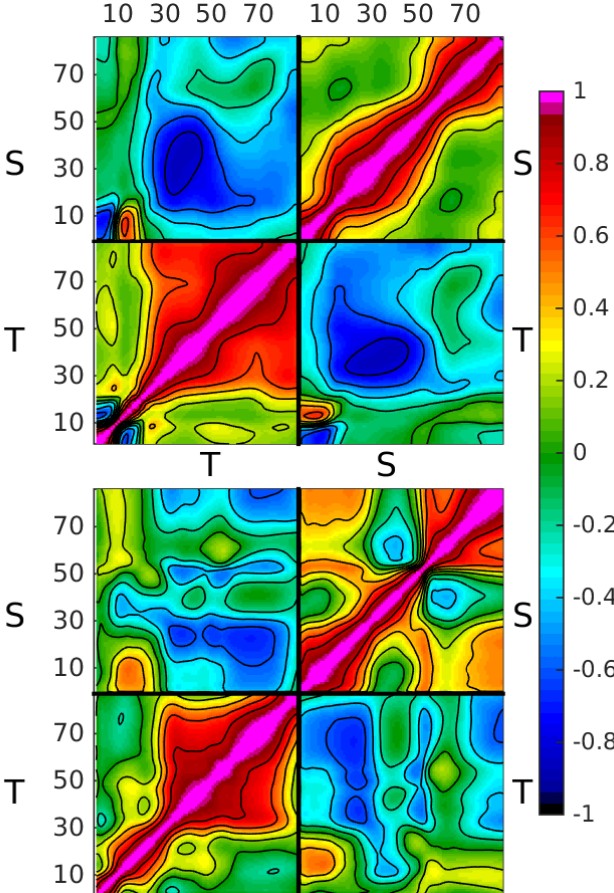

**Figure 6: The top Panel shows the static and spatially homogeneous vertical error correlation matrix, the bottom panel the ensemble estimate on 22 June 2014 at lat 7.0°N and lon 40.0°E. The numbers on the axis indicate to the model levels where the first 78 values represent temperature and the second 78 values represent the salinity levels. The matrix blocks represent the correlation of T and S and their cross-correlations.**

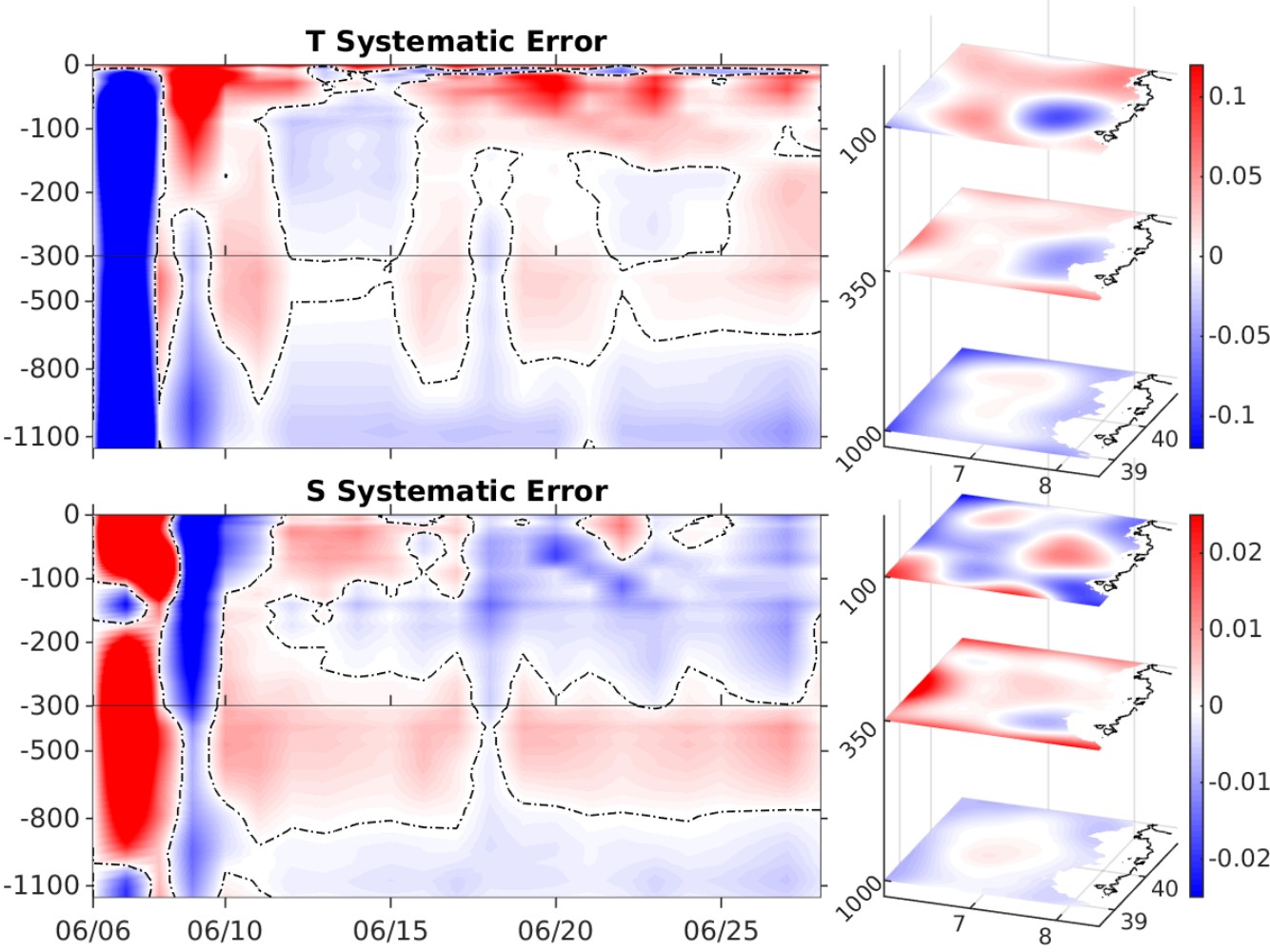

**Figure 7: Left panels: Y axis indicates depth in meter, X axis indicates time in days, colour are the Systematic error correction horizzontally averaged over the whole model domain for temperature (°C, top) and salinity (bottom). Right panels: Z axis indicates depth, X and Y are latitude and longitude respectively, the colours are the Systematic error correction averaged between 12 and 28 Jun at 100m, 350m and 1000m detph for Temperature (°C, top) and Salinity (bottom).**

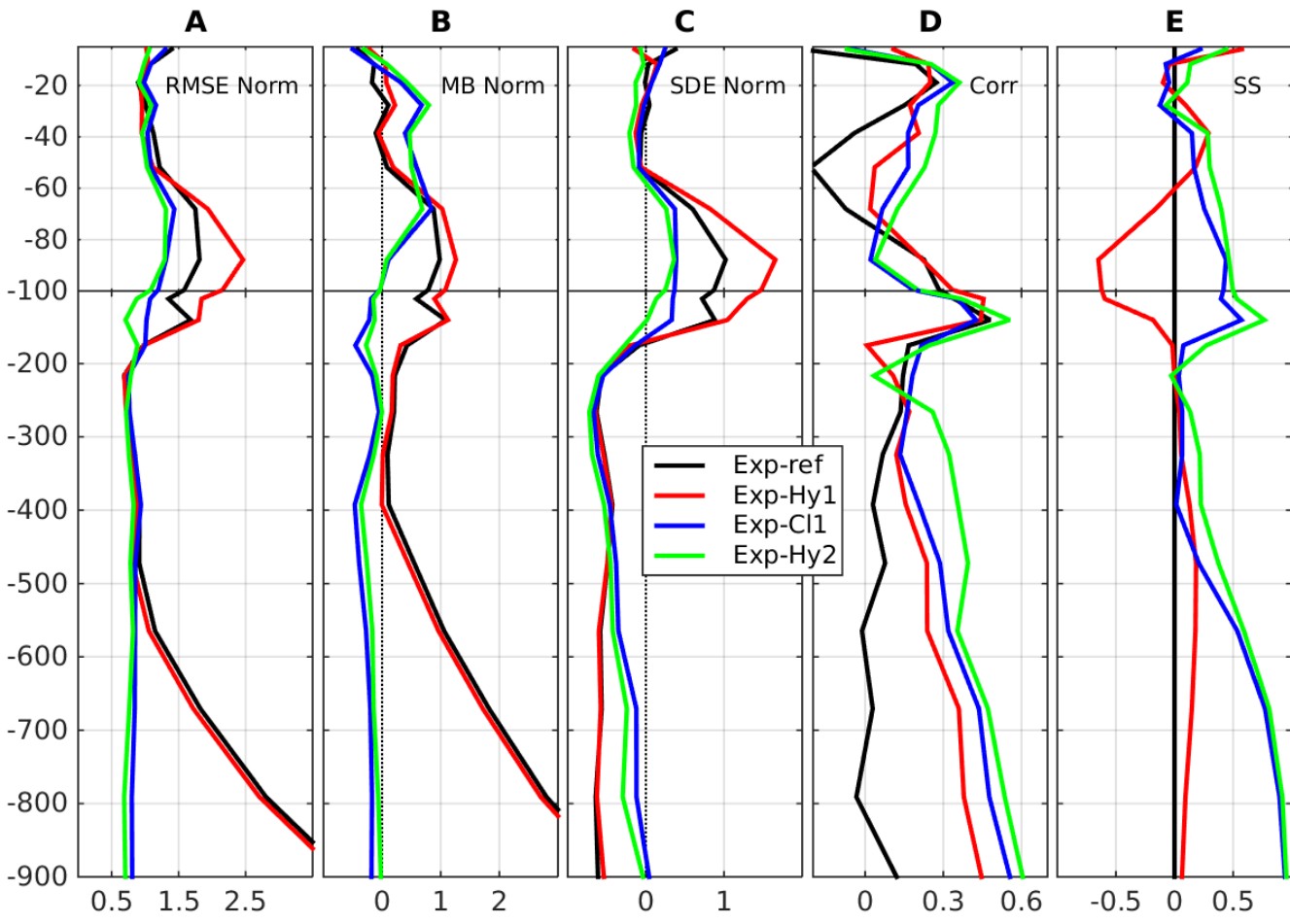

**Figure 8: Vertical profiles of temperature error components and Skill Score for the different experiments: black lines indicate the Exp-ref results; red lines indicate Exp-Hy1 results; blue lines indicate Exp-Cl1 results and green lines indicate Exp-Hy2 results. A: normalized Root Mean Square Error. B: normalized Mean Bias. C: Normalized Standard Deviation Error. D: Cross-Correlation. E: Skill Score.**

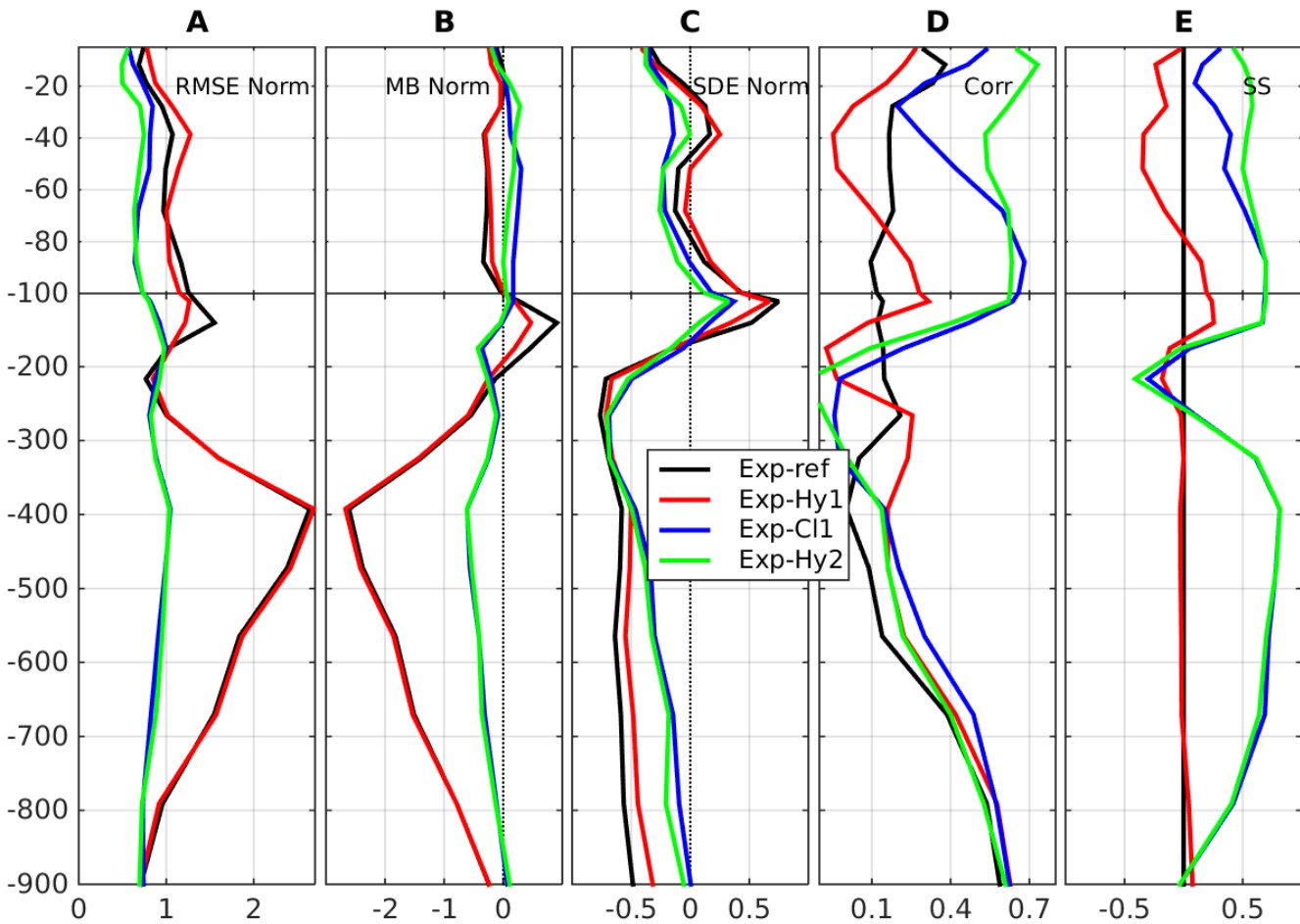

**Figure 9: As Figure 8 but for Salinity.**