# Peer review of "A Hybrid Variational-Ensemble data assimilation scheme with systematic error correction for limited area ocean models"

_Ocean Science, 2016_

## Referee Comment (RC1) · P. Sakov (Referee) · 27 Jun 2016

[11pt]article natbib amsmath amssymb [breaklinks]hyperref

**1   General comments**

The manuscript investigates application of a hybrid variational-ensemble approach to a limited area model with sub-mesoscale resolution. So far constraining sub-mesoscale features remains largely out of reach of contemporary ocean forecasting with the standard set of remote and in-situ observations. This study investigates sub-mesoscale

forecasting in situation with dense in-situ observations in a relatively calm region.

The hybrid approaches become popular in atmospheric and ocean forecasting due to increasingly clear understanding of limitations of 4D-Var systems due to lack of mechanisms to carry information forward from previous cycles. In my view, the hybrid approaches, while succeeding in adding some degree of flow dependence to the background covariance, still remain largely empirical and lack consistent formulation. As a consequence, to the best of my knowledge, there still no published experiments with small models that would convincingly demonstrate advantages of hybrid systems over much more simple and consistent EnKF systems.

Below I will list some major and minor issues I see with the manuscript and give recommendation in the conclusion.

**2  Major Issues**

1. **Equations (6), (7) and (8).**

   One problem with mixing covariance matrices is that there is no good way to incorporate it in a consistent way into the optimisation problem. In particular, the claim that linear mixing (6) can be consistent with the cost function (8) is generally wrong. The framework (6-8) assumes $(\mathbf{x}_c + \mathbf{x}_e)^{\mathsf{T}} [\alpha \mathbf{B}_c + (1-\alpha)\mathbf{B}_e]^{-1} (\mathbf{x}_c + \mathbf{x}_e)$ $= (\mathbf{x}_c)^{\mathsf{T}} (\alpha \mathbf{B}_c)^{-1} \mathbf{x}_c + (\mathbf{x}_e)^{\mathsf{T}} [(1-\alpha)\mathbf{B}_e]^{-1} \mathbf{x}_e$. This implies $\mathbf{x}_c \perp \mathbf{x}_e$, $\mathbf{B}_c \perp \mathbf{B}_e$, which is generally not true and difficult (or even impossible) to impose in practice. (The same applies to Eqs. 4-6 in Wang et al. 2007.)

   If the above is true, then the manuscript must be modified accordingly.

2. **Equation (14).**

   It seems to me that it writes "innovation = innovation error", which is wrong.
3. **Ensemble update.**

   Due to the lack of rigorous formulation most hybrid methods employ empirical approaches for maintaining the ensemble spread. It seems that the manuscript does not tell explicitly how the ensemble members are updated. This is important for understanding the method and should be described.

   Further, on p. 10, l. 12-23 it is stated that the ensemble maintains spread due to observations and otherwise collapses due to the deterministic model forcing. This is somewhat contrary to what might be expected. It seems to me that increasing the number of observations in a consistent DA system should *always* reduce state error, that is always reduce the ensemble spread. Concerning the model forcing, in the context of a mainly stable forcing-driven model it is probably a pre-requisite to perturb forcing for ensemble members to match the corresponding uncertainty.

**3 Minor Issues**

1. P. 2, l. 1: suggest replacing "not feasible to sample" by "not feasible to observe".

2. P. 2, l. 24: suggest replacing "EnKF" by "traditional EnKF".

3. P. 6, l. 19: suggest replacing "model bias error" by "model bias".

4. P. 7, l. 3: suggest replacing "background error covariances" by "background errors".

**4 Conclusion and recommendations**

The manuscript addresses a difficult and interesting ocean forecasting problem. Some of the statements and approaches can be viewed as arguable (or, in regard to the
EnKF, outdated), but this is what scientific literature is for. The methods used are in my view largely empirical, and again there is nothing wrong with that, as long this is clearly stated up-front.

Concerning the results of the DA experiment described, they probably leave a lot of space for improvement, and this itself is one of the important outcomes of the manuscript.

One line of statements that I tend to disagree with is that "it is difficult to run full EnKF with a large number of members" (p. 3, l. 14-17). Not in the year 2016, and definitely not with a 240 x 240 x 90 model.

Overall, I believe that the manuscript will be interesting and useful for the ocean modelling and ocean forecasting communities. I recommend publishing it in Ocean Science after fixing the major issues listed above, which probably amounts to a major revision.

**References**

Wang, X., C. Snyder, and T. M. Hamill, 2007: On the theoretical equivalence of differently proposed ensemble-3DVAR hybrid analysis schemes. *Mon. Wea. Rev.*, **135**, 222–227.

---

## Referee Comment (RC2) · Anonymous Referee #2 · 7 Jul 2016

This paper presents an assessment of a hybrid variational-ensemble scheme and looks at the impact of systematic error correction and the inclusion of systematic bias correction. I generally found this a good and interesting paper. I do have a few mostly minor comments which I'll list below along with the page and line number.

page 1 title. Should be either "error correction for a limited area ocean model" or "error correction for limited area ocean models"

page 1 line 23-25. Slightly confusing sentence. What's "hybrid daily estimates" perhaps this would read better as "and a hybrid of the background error covariance ... ensemble derived errors of the day" or similar

page 1 line 30. present day observations.

page 2 line 10. Divergence happens in chaotic systems even if the model is perfect it's not just a result of model bias.

page 3 line 26. Write "A promising application..." Also it would be useful to be more specific about the findings of Penny et al.

page 3 line 33. I suppose you mean having a long lengthscale for the bias is more straightforward to implement than implementing multiple lengthscales for the control variables.

page 4 line 8. independent on the -> independent of the

page 4 equations onwards. Inconsistent bolding of vectors. Check the journal style but I think you should use bold roman for vectors and bold roman capitals for Matrices.

page 5 line 16. Say a bit more about how this adjustment is made.

page 6 line 4. You don't use the ensemble to estimate the horizontal lengthscales here. You should say so to avoid confusion

page 6 line 21. I'm not sure I agree that inaccurate initial conditions produce a systematic error or bias. Initial condition error is likely to average to zero over time and is not therefore systematic.

page 6 line 25. "This idea ... " I don't understand this sentence. Can you rewrite it.

page 7 line 4. Should be "It is worth mentioning that the..."

page 7 line 7. Should be "simulation allows us to retrieve..."

page 7 line 21. This is true if there is no bias in the observations.

page 7 line 30. Add brackets something like min(J(dx))

page 8 line 12 "Sardinia has been conducted" -> "Sardinia was conducted"

page 8 line 17 "accounting for" -> by. Remove "data"

page 8 line 19. "remote sensing" -> "remotely sensed"

page 8 line 28. "Fig.01" -> Fig. 1 and similar elsewhere.

page 8 line 32. "mean" -> "means"

page 9. line 1-10. I found the perturbation of the observations confusing and not well explained. Are you vertically subsampling the profiles? Please clarify the text.

page 10. ~Line 20. I wonder if you need to perturb things other than the observations and following on from my previous comment are you perturbing the observations enough. See also p 14. Surely it's better to perturb everything even is the effect is limited by the short integration time.

page 11. Line 31. Remove "biases in"

page 12. Line 16. "while goes" -> "while it goes"

page 12. Line 18. "that also" -> "also that"

page 12. Line 25. "Mean" -> "mean"

page 13. Are you statistics computed after or before assimilation is it observations compared to analysis or observations compared to model background. Or are the CTD data used for comparison independent? It's not clear.

page 13. It might be worth not abbreviating in all cases as it makes it more difficult to read. For example MB perhaps just write mean bias. Similarly with SS = skill score.

page 13. Line 13-14 "capable of significantly reducing this bias (error)". Perhaps remove the error?

page 13. Line 32. Remove "can"

page 15. Line 18. Typo "indicating"

page 16. Line 2 "of" -> "by"

page 16. Line 10. Do you plan to use this Baysesian method?

page 16. References - inconsistent "Mon. Wea. Rev."

page 17. Reference Mirouze has a typo "LOCKLEY" in the author list.

Table 3. Why is the 0-50 mean bias worse in the bias corrected runs?

Table 3. Not enough significant figures to see anything useful. I think it would more useful to use MB and SDE rather than the squares. Either that or add a significant figure.

Figure 1. Could say the date range over which the observations are plotted.

Figure 2. Would be good to add a legend to this plot and some axis labels.

Figure 3. Don't understand the explanation of the middle panel. Label x axis

Figure 5. Spread == standard deviation of ensemble ?

Figure 6. Are both matrices are for the same location?

Figure 7. Give the depths of the horizontal slices. Typos in the caption.

Figure 8 (also Figure 9). Consider plotting MB and SDE rather than the squares it will make it easier to distinguish the lines particularly where the errors are lower. A legend would be useful on this plot too.

Figure 8 (also Figure 9). I notice that the non-hybrid results are quite good sometimes better than the hybrid results why might this be? It may be a case of needing to do more tuning perhaps and that not making it worse is quite a good result perhaps. It might be worth saying a bit more about this in the text.

---

## Author Response (AR1)

**Comment from Referee#1: 1 General comments**

The manuscript investigates application of a hybrid variational-ensemble approach to a limited area model with sub-mesoscale resolution. So far constraining sub-mesoscale features remains largely out of reach of contemporary ocean forecasting with the standard set of remote and in-situ observations. This study investigates sub-mesoscale forecasting in situation with dense in-situ observations in a relatively calm region. The hybrid approaches become popular in atmospheric and ocean forecasting due to increasingly clear understanding of limitations of 4D-Var systems due to lack of mechanisms to carry information forward from previous cycles. In my view, the hybrid approaches, while succeeding in adding some degree of flow dependence to the background covariance, still remain largely empirical and lack consistent formulation. As a consequence, to the best of my knowledge, there still no published experiments with small models that would convincingly demonstrate advantages of hybrid systems over much more simple and consistent EnKF systems. Below I will list some major and minor issues I see with the manuscript and give recommendation in the conclusion.

*Author's response:*

*We thanks Dr. P Sakov for carefully reading our manuscript and the useful comments he provided. We completely agree on the major challenges we try to address that Dr. Sakov has correctly identified, as well as strengthens and weakness of the proposed approach. In particular, we agree that part of the methodology is somehow empirical, although largely adopted, and we will better specify this point in the revised version of the manuscript.*

**Comment from Referee: 2 Major Issues**

1. Equations (6), (7) and (8).

One problem with mixing covariance matrices is that there is no good way to incorporate it in a consistent way into the optimisation problem. In particular, the claim that linear mixing (6) can be consistent with the cost function (8) is generally wrong. The framework (6-8) assumes

$$(\mathbf{x}_c+\mathbf{x}_e)^T[\alpha \boldsymbol{B}_c + (1-\alpha)\boldsymbol{B}_e]^{-1}(\mathbf{x}_c+\mathbf{x}_e) = (\mathbf{x}_c)^T(\alpha \boldsymbol{B}_c)^{-1}\mathbf{x}_c + (\mathbf{x}_e)^T[(1-\alpha)\boldsymbol{B}_e]^{-1}\mathbf{x}_e$$

This implies $x_c \perp x_e$, $B_c \perp B_e$; which is generally not true and difficult (or even impossible) to impose in practice. (The same applies to Eqs. 4-6 in Wang et al. 2007.) If the above is true, then the manuscript must be modified accordingly.

*Author's response:*

*We thank Dr. Sokov for this comment, but we only partially agree. The equality written by the reviewer does require the orthogonality. On the other hand, in order to use eq.8 instead of eq.6, it is only necessary that the cost function in eq.8 reaches the minimum for the same values of δx as the one in eq.6 and the ensemble generated by the analysis using eq.8 estimates the same covariance matrix as the one using eq.6. To demonstrate it we only need to assume that in eq.8 $x_c$ and $x_e$, may be perturbed independently and that **B** contains the true background-error covariances, i.e. the background errors are well specified. These assumptions are much weaker than imposing the orthogonality on both the $x_c$,*

$x_e$ and $B_c$, $B_e$. *We think that the independency between $x_c$ and $x_e$ is a reasonable assumption since the two variables separately sample historical events and current forecast, while the suitability of* **B** *implicitly relies on the quality of the ensemble and climatological error estimates. We will add the following Appendix in the manuscript with the full derivation of eq.8.*

***Author's changes in manuscript:***

***APPENDIX***

*We start from the cost function:*

$$J(\delta x) = \frac{1}{2}\delta x^T B^{-1}\delta x + \frac{1}{2}(H\delta x - d)^T R^{-1}(H\delta x - d) \tag{A.1}$$

*To define our hybrid assimilation schemes we compute* **B** *as a linear combination of the "static" covariance operator,* **B**$_c$*, and the flow-dependent operator,* **B**$_e$*:*

$$B = \alpha B_c + (1-\alpha)B_e \tag{A.2}$$

*where $\alpha$ is the relative weight. Substituting A.2 in A.1 we obtain the new hybrid cost function:*

$$J(\delta x) = \frac{1}{2}\delta x^T (\alpha B_c + (1-\alpha)B_e)^{-1}\delta x + \frac{1}{2}(H\delta x - d)^T R^{-1}(H\delta x - d) \tag{A.3}$$

*We define now the increment as a weighted sum of parts corresponding to static and flow-dependent covariance matrices:*

$$\delta x = \delta x_c + \delta x_e.$$

*We want to demonstrate that:*

$$J(\delta x) = \frac{1}{2}\delta x_c^T (\alpha B_c)^{-1}\delta x_c + \frac{1}{2}\delta x_e^T ((1-\alpha)B_e)^{-1}\delta x_e + \frac{1}{2}(H\delta x - d)^T R^{-1}(H\delta x - d) \tag{A.4}$$

*has the minimum for the same value of $\delta x$ as A.3.*

*To minimize A.4, $\delta x_c$ and $\delta x_e$ must satisfy $\frac{\partial J(\delta x)}{\partial x_c} = 0$ and $\frac{\partial J(\delta x)}{\partial x_e} = 0$ which gives:*

$$(\alpha B_c)^{-1}\delta x_c + \frac{\partial}{\partial x_c}\left(\frac{1}{2}\delta x_e^T ((1-\alpha)B_e)^{-1}\delta x_e\right) + \frac{1}{2}\frac{\partial J_o}{\partial x_c} = 0 \tag{A.5}$$

$$[(1-\alpha)B_e]^{-1}\delta x_e + \frac{\partial}{\partial x_e}\left(\frac{1}{2}\delta x_c^T (\alpha B_c)^{-1}\delta x_c\right) + \frac{1}{2}\frac{\partial J_o}{\partial x_e} = 0 \tag{A.6}$$

*where Jo is the observational term. Assuming that $\delta x_c$ and $\delta x_e$ can be perturbed independently, both the second terms on the left hand side of A.5 and A.6 are null:*

$$\frac{\partial}{\partial x_c}\left(\frac{1}{2}\delta x_e^T ((1-\alpha)B_e)^{-1}\delta x_e\right) = 0, \tag{A.7}$$

$$\frac{\partial}{\partial x_e}\left(\frac{1}{2}\delta x_c^T (\alpha B_c)^{-1}\delta x_c\right) = 0 \tag{A.8}$$

*and*

$$\frac{\partial J_o}{\partial x} = \frac{\partial J_o}{\partial x_e} = \frac{\partial J_o}{\partial x_c} = 2H^T R^{-1}(H\delta x - d). \tag{A.9}$$

*This is a reasonable assumption, because the two random values are sampled from different Gaussians. Although they are defined over the same space, one is sampled from historical states, and the other from current forecasts. Premultiplying A.5 by $\alpha B_c$ and A.6 by $(1-\alpha)B_e$, removing the null terms, summing the two subsequent equations and applying A.9 yields:*

$$0 = (\delta x_c + \delta x_e) + \frac{1}{2}[\alpha \mathbf{B}_c + (1-\alpha)\mathbf{B}_e)]\frac{\partial J_o}{\partial x} \tag{A.10}$$

*Multiplying A.10 by the inverse of the hybrid covariance:*

$$0 = [\alpha \mathbf{B}_c + (1-\alpha)\mathbf{B}_e)]^{-1}(\delta x_c + \delta x_e) + \mathbf{H}^T\mathbf{R}^{-1}[\mathbf{H}(\delta x_c + \delta x_e) - \mathbf{d}] \tag{A.11}$$

*This is also the minimum of A.3 that we wanted as a proof.*

*Furthermore, defining the background and analysis perturbations around the true state $x_t$ as:*

$$\delta x_b = x_b - x_t$$

*and*

$$\delta x_a = x_a - x_t,$$

*by adding and subtracting the true state A.11 becomes:*

$$0 = [\alpha \mathbf{B}_c + (1-\alpha)\mathbf{B}_e)]^{-1}(x_a - x_b - x_t + x_t) + \mathbf{H}^T\mathbf{R}^{-1}[\mathbf{H}(x_a - x_b - x_t + x_t) - (y - Hx_b)]$$

*or:*

$$0 = [\alpha \mathbf{B}_c + (1-\alpha)\mathbf{B}_e)]^{-1}(\delta x_a - \delta x_b) + \mathbf{H}^T\mathbf{R}^{-1}[\mathbf{H}\delta x_a - (y - Hx_t)]$$

*that can be written also as:*

$$\{[\alpha \mathbf{B}_c + (1-\alpha)\mathbf{B}_e)]^{-1} + \mathbf{H}^T\mathbf{R}^{-1}\mathbf{H}\}\delta x_a = [\alpha \mathbf{B}_c + (1-\alpha)\mathbf{B}_e)]^{-1}\delta x_b + \mathbf{H}^T\mathbf{R}^{-1}[y - Hx_t]. \text{A.12}$$

*Multiplying each side of A.12 by its transpose, taking the expectation, assuming that observational errors are independent of background errors:*

$$\{[\alpha \mathbf{B}_c + (1-\alpha)\mathbf{B}_e)]^{-1} + \mathbf{H}^T\mathbf{R}^{-1}\mathbf{H}\}A\{[\alpha \mathbf{B}_c + (1-\alpha)\mathbf{B}_e)]^{-1} + \mathbf{H}^T\mathbf{R}^{-1}\mathbf{H}\}^T = [\alpha \mathbf{B}_c + (1-\alpha)\mathbf{B}_e)]^{-1}E\{\delta x_b(\delta x_b)^T\}[\alpha \mathbf{B}_c + (1-\alpha)\mathbf{B}_e)]^{-T} + \mathbf{H}^T\mathbf{R}^{-1}\mathbf{R}\mathbf{R}^{-1}\mathbf{H}. \tag{A.13}$$

*Assuming the **B** contains the true background error covariances, i.e. the background errors are well specified, and using A.2:*

$$E\{\delta x_b(\delta x_b)^T\} = \alpha \mathbf{B}_c + (1-\alpha)\mathbf{B}_e$$

*thus:*

$$\{[\alpha \mathbf{B}_c + (1-\alpha)\mathbf{B}_e)]^{-1} + \mathbf{H}^T\mathbf{R}^{-1}\mathbf{H}\}A\{[\alpha \mathbf{B}_c + (1-\alpha)\mathbf{B}_e)]^{-1} + \mathbf{H}^T\mathbf{R}^{-1}\mathbf{H}\}^T = [\alpha \mathbf{B}_c + (1-\alpha)\mathbf{B}_e)]^{-1}[\alpha \mathbf{B}_c + (1-\alpha)\mathbf{B}_e)][\alpha \mathbf{B}_c + (1-\alpha)\mathbf{B}_e)]^{-T} + \mathbf{H}^T\mathbf{R}^{-1}\mathbf{R}\mathbf{R}^{-1}\mathbf{H}. \tag{A.14}$$

*or :*

$$\{[\alpha \mathbf{B}_c + (1-\alpha)\mathbf{B}_e)]^{-1} + \mathbf{H}^T\mathbf{R}^{-1}\mathbf{H}\}A\{[\alpha \mathbf{B}_c + (1-\alpha)\mathbf{B}_e)]^{-1} + \mathbf{H}^T\mathbf{R}^{-1}\mathbf{H}\}^T$$
$$= [\alpha \mathbf{B}_c + (1-\alpha)\mathbf{B}_e)]^{-1} + \mathbf{H}^T\mathbf{R}^{-1}\mathbf{H}.$$

*Dividing by $\{[\alpha \mathbf{B}_c + (1-\alpha)\mathbf{B}_e)]^{-1} + \mathbf{H}^T\mathbf{R}^{-1}\mathbf{H}\}$:*

$$A = \{[\alpha \mathbf{B}_c + (1-\alpha)\mathbf{B}_e)]^{-1} + \mathbf{H}^T\mathbf{R}^{-1}\mathbf{H}\}^{-1} \tag{A.15}$$

*where $A = E\{\delta x_a(\delta x_a)^T\}$ is the analysis error covariance matrix. A.15 demonstrates that independent forecasts updates in each ensemble member by using A.4 give the same optimal estimate of updated covariances as A.3.*

**Comment from Referee: 2. Equation (14).**

It seems to me that it writes "innovation = innovation error", which is wrong.

***Author's response:***

*We thank the reviewer for this comment. We think our formulation is correct as the following is valid. Our eq.14 in the manuscript is:*

$$d = [y - H(x_b)] = \varepsilon_o - (\varepsilon_r + \varepsilon_s)$$

*where $d$ is the misfit, $\varepsilon_o$ is the observational error, $\varepsilon_r$ is the background random error and $\varepsilon_s$ is the background systematic error.*

*Introducing the true state of the ocean $x_t$, (14) can be written also as:*

$$d = [y - H(x_b)] = y - H(x_t) + H(x_t) - H(x_b) = \varepsilon_o - H(\varepsilon_r + \varepsilon_s)$$

*where the errors are defined as departures from the true state. If the observation network is dense, H ~ I, reducing to Eq. (14).*

***Author's changes in manuscript:***

*We will modify eq.14 in the manuscript including the intermediate equivalence to clarify.*

**Comment from Referee: 3. Ensemble update.**

Due to the lack of rigorous formulation most hybrid methods employ empirical approaches for maintaining the ensemble spread. It seems that the manuscript does not tell explicitly how the ensemble members are updated. This is important for understanding the method and should be described. Further, on p. 10, l. 12-23 it is stated that the ensemble maintains spread due to observations and otherwise collapses due to the deterministic model forcing. This is somewhat contrary to what might be expected. It seems to me that increasing the number of observations in a consistent DA system should always reduce state error, that is always reduce the ensemble spread. Concerning the model forcing, in the context of a mainly stable forcing-driven model it is probably a pre-requisite to perturb forcing for ensemble members to match the corresponding uncertainty.

***Author's response:***

*We fully agree that in a classical DA system increasing the number of observations should reduce the ensemble spread. This is due to the ensemble members' generation procedure that usually involve perturbation in the initialization, surface and lateral open boundary conditions and in model physics (e.g. unresolved scales) as well.*

*In our experiment, differences between the ensemble members are generated perturbing only the observations, as explained in the manuscript p. 8 line 29 to page 9 line 13.*

*For the time being initialization, atmospheric forcing and lateral open boundary condition are unperturbed, the ensemble generation method spans the uncertainty linked with the observational sampling and assimilation formulation, implicitly acting on the background ensemble spread. This approach implicitly relies on a perfect model assumption, and it is likely to under-estimate the ensemble covariances, implying that when no observations are assimilated, the spread equals 0 by construction, unlike most ensemble systems with full perturbations. By not perturbing the surface and lateral boundary conditions, we assume that the flow-dependent component of **B** is associated with the small-scale error fluctuations. Thus the deterministic large-scale forcing acts as an attractor for the ensemble*

*perturbations, especially at the sea surface and in proximity of the boundaries. We plan in the future to relax this assumption and introduce of full set of perturbations spanning most of the uncertainties in the system.*

*During the experiment we have assimilated a total of 3139 temperature and salinity vertical profiles deriving from gliders and CTD stations (255 CTD and 2884 Gliders vertical profiles). The perturbation of the observations, which is one the empirical aspects mentioned by the reviewer in his general comment of the manuscript, produces sensible differences in the observations and through differences in the solution of the minimization propagates in the model background of the following assimilation cycle. Each of the 14 ensemble members is an independent simulation with its own perturbation function and associated horizontal correlation radius, every assimilation cycle the observations differ but also the background of the individuals members are different, this produces the ensemble spread discussed and illustrated in Fig.4 and Fig.5. This method clearly strongly connects the growth of the ensemble spread to the perturbation function used and simultaneously link the ensemble spread to observations availability being otherwise forced by the same deterministic conditions (lateral and surface open boundaries).*

***Author's changes in manuscript:***

*We will include and mention more explicitly these issues in the revised version of the manuscript.*

**Comment from Referee: 3 Minor Issues**

1. P. 2, l. 1: suggest replacing "not feasible to sample" by "not feasible to observe".

2. P. 2, l. 24: suggest replacing "EnKF" by "traditional EnKF".

3. P. 6, l. 19: suggest replacing "model bias error" by "model bias".

4. P. 7, l. 3: suggest replacing "background error covariances" by "background errors".

***Author's response:***

*We thank the reviewer for these suggestions.*

***Author's changes in manuscript:***

*The revised version of the manuscript will be modified accordingly.*

**Comment from Referee: 4 Conclusion and recommendations**

The manuscript addresses a difficult and interesting ocean forecasting problem. Some of the statements and approaches can be viewed as arguable (or, in regard to the EnKF, outdated), but this is what scientific literature is for. The methods used are in my view largely empirical, and again there is nothing wrong with that, as long this is clearly stated up-front. Concerning the results of the DA experiment described, they probably leave a lot of space for improvement, and this itself is one of the important outcomes of the manuscript. One line of statements that I tend to disagree with is that "it is difficult to run full EnKF with a large number of members" (p. 3, l. 14-17). Not in the year 2016, and definitely not with a 240 x 240 x 90 model. Overall, I believe that the manuscript will be interesting and useful

for the ocean modelling and ocean forecasting communities. I recommend publishing it in Ocean Science after fixing the major issues listed above, which probably amounts to a major revision.

*Author's response:*

*We thank the Reviewer for the general positive comments on the manuscript. In the revised version we tried to address all the major comments provided. Concerning the statements of p3. Line 14-17, this is written in the introduction and it is not explicitly linked to our experiment but rather to operational oceanography applications, however we will rephrase the sentence.*

*Author's changes in manuscript:*

*p3. Line 14-17 "Furthermore present computational resources limit the number of ensemble members accounted in operational EnKF."*

**References**

Wang, X., C. Snyder, and T. M. Hamill, 2007: On the theoretical equivalence of differently proposed ensemble-3DVAR hybrid analysis schemes. Mon. Wea. Rev., 135, 222–227.

**Comment from Referee#2: General comments**

This paper presents an assessment of a hybrid variational-ensemble scheme and looks at the impact of systematic error correction and the inclusion of systematic bias correction. I generally found this a good and interesting paper. I do have a few mostly minor comments which I'll list below along with the page and line number.

*Author's response:*

*We thank the Reviewers for his/her positive comments. Below we provide answers to each comment.*

**Comment from Referee#2:**

−       page 1 title. Should be either "error correction for a limited area ocean model" or "error correction for limited area ocean models"

−       page 1 line 23-25. Slightly confusing sentence. What's "hybrid daily estimates" perhaps this would read better as "and a hybrid of the background error covariance ... ensemble derived errors of the day" or similar

−       page 1 line 30. present day observations.

*Author's response: We Thank the Reviewer for his/her comments.*

*Author's changes in manuscript: All the suggestions will be implemented in the revised version of the manuscript.*

**Comment from Referee#2:** page 2 line 10. Divergence happens in chaotic systems even if the model is perfect it's not just a result of model bias.

*Author's response: We Thank the Reviewer for his/her comments.*

*Author's changes in manuscript: We agree with the Reviewers and the sentence will be modified as follows:*

*"These approximations affect the model solutions in terms of quality and accuracy and, more importantly, differences between the numerical solution and the true state amplify along time due to the chaotic component of the ocean dynamic."*

**Comment from Referee#2:** page 3 line 26. Write "A promising application..." Also it would be useful to be more specific about the findings of Penny et al.

*Author's response: We Thank the Reviewer for his/her comments.*

*Author's changes in manuscript: In the revised version of the manuscript we will include the following sentence:*

*"They compared hybrid, classical 3DVAR and EnKF schemes in an observing system simulation experiment and using also real data, showing that the hybrid scheme reduces errors for all prognostic model variables eliminating growth in biases present in the EnKF and 3DVAR."*

**Comment from Referee#2:** page 3 line 33. I suppose you mean having a long length scale for the bias is more straightforward to implement than implementing multiple length scales for the control variables.

*Author's response: We Thank the Reviewer for his/her comments.*

*Author's changes in manuscript: The sentence will be rephrased as follows:*

*"A possible simplification is to assume that systematic errors are characterized by long length scales, as often occurs to some extent (Dee 2005)."*

**Comment from Referee#2:** page 4 line 8. independent on the -> independent of the

*Author's response: Thank, the revised version of the manuscript will be modified accordingly.*

**Comment from Referee#2:** page 4 equations onwards. Inconsistent bolding of vectors. Check the journal style but I think you should use bold roman for vectors and bold roman capitals for Matrices.

*Author's response:* We thank the Reviewers for this comments. According Journal style Matrices are printed in boldface, and vectors in boldface italics.

*Author's changes in manuscript:* we will modify the manuscript accordingly.

**Comment from Referee#2:** page 5 line 16. Say a bit more about how this adjustment is made.

*Author's response:* We thank the reviewers, the following explanation will be included in the revised version of the manuscript just after eq.6:

*Author's changes in manuscript:*

"The relative weighting (α) still requires empirical tuning but in general can be adjusted to the size of the ensemble. Large ensemble size can provide robust estimate of $\mathbf{B_e}$ and thus 6 can be theoretically implemented with small α values (Menetrier and Auligne', 2015)."

**Comment from Referee#2:** page 6 line 4. You don't use the ensemble to estimate the horizontal length scales here. You should say so to avoid confusion

*Author's response:* In the experiment with hybrid covariance matrices we use the ensemble to estimate also the horizontal length scales varying daily. This is stated in Section 3 "Experimental set-up" page 9 line 27 and Table 2 of the original version of the manuscript.

*Author's changes in manuscript:* We suggest that the text remains unchanged in the revised version of the manuscript.

**Comment from Referee#2:** page 6 line 21. I'm not sure I agree that inaccurate initial conditions produce a systematic error or bias. Initial condition error is likely to average to zero over time and is not therefore systematic.

*Author's response:* We thank the Reviewers for this comment. There may be confusion: for initial condition we meant the climatological state we start the experiments from (i.e. the system initialization).We think that the temporal scales considered play a major role. Initial condition error will average to zero if the system is integrated for enough time.

*Author's changes in manuscript:* We will modify the sentence in order to avoid confusion.

**Comment from Referee#2:** page 6 line 25. "This idea ... " I don't understand this sentence. Can you rewrite it.

*Author's response:* Thank, we will rephrase the sentence.

*Author's changes in manuscript:*

*"This idea is consistent with the high-resolution model presented in Section 3 and with the experimental setup where the large scale uncertainties (initialization, boundary conditions and surface forcing) are not accounted in the generation of the ensemble members."*

**Comment from Referee#2:**

−        page 7 line 4. Should be "It is worth mentioning that the..."

−        page 7 line 7. Should be "simulation allows us to retrieve..."

*Author's response: Thank, the revised version of the manuscript will be modified accordingly.*

**Comment from Referee#2:** page 7 line 21. This is true if there is no bias in the observations.

*Author's response: We agree with the reviewer, however this is explicitly mentioned in the original version of the manuscript at line 19: "assuming that also the observational error is unbiased".*

*Author's changes in manuscript:  We suggest not to modify the manuscript.*

**Comment from Referee#2:**

−        page 7 line 30. Add brackets something like min(J(dx))

−        page 8 line 12 "Sardinia has been conducted" -> "Sardinia was conducted"

−        page 8 line 17 "accounting for" -> by. Remove "data"

−        page 8 line 19. "remote sensing" -> "remotely sensed"

−        page 8 line 28. "Fig.01" -> Fig. 1 and similar elsewhere.

−        page 8 line 32. "mean" -> "means"

*Author's response: Thank, the revised version of the manuscript will be modified accordingly.*

**Comment from Referee#2:** page 9. line 1-10. I found the perturbation of the observations confusing and not well explained. Are you vertically subsampling the profiles? Please clarify the text.

*Author's response: Thank, the revised version of the manuscript will be modified accordingly.*

*Author's changes in manuscript:  We will rephrase the explanation as follows:*

*"The ensemble members have been generated simultaneously assimilating perturbed observations, varying the corresponding observational error, and assuming different horizontal correlation radii in $V_H$. For the observation perturbation, either weak or strong criteria for retaining observations are used among the ensemble members, where strong*

*quality check procedure requires both temperature and salinity observations are flagged as good, reducing the total number of assimilated observations. Filters have been applied horizontally and vertically to reduce the higher spatial sampling of observations with respect to the model grid. Within the ensemble members, different vertical cut-off scales have been used in a low pass filter, resulting in differently smoothed profiles. Horizontally, data binning has been applied to the observations falling in 1 or 2 model grid cells while keeping the original vertical resolution. When the filtering or binning procedures are applied, the corresponding full resolution profile standard deviation has been used to as an estimate of the observational error. Similar procedures have been applied to CTD and Gliders data."*

**Comment from Referee#2:** page 10. Line 20. I wonder if you need to perturb things other than the observations and following on from my previous comment are you perturbing the observations enough. See also p 14. Surely it's better to perturb everything even is the effect is limited by the short integration time.

*Author's response: Generating the ensemble members only by perturbing the observations clearly poses the constraint of observations availability. The absence of observations reduces significantly the ensemble spread. As discussed in the manuscript (page 10 lines 22, 23) the methodology can be improved including perturbations in the initialization and in the lateral or surface boundary conditions. However the perturbation of initial, lateral and surface boundary condition could create overlaps between static-climatological background-error covariances, daily-varying ensemble-derived covariances, and large-scale systematic error corrections. As first step we preferred to keep a conservative approach ensuring the separations of the two scales.*

*Author's changes in manuscript:* *We think that changes in the manuscript due to other comments will better clarify this issue.*

**Comment from Referee#2:**

−      page 11. Line 31. Remove "biases in"

−      page 12. Line 16. "while goes" -> "while it goes"

−      page 12. Line 18. "that also" -> "also that"

−      page 12. Line 25. "Mean" -> "mean"

*Author's response: Thank, the revised version of the manuscript will be modified accordingly.*

**Comment from Referee#2:** page 13. Are you statistics computed after or before assimilation is it observations compared to analysis or observations compared to model background. Or are the CTD data used for comparison independent? It's not clear.

*Author's response: All the statistics are computed using the model background, i.e. before data are assimilated. We thank the reviewer for pointing this out.*

*Author's changes in manuscript: We will specify the dataset used in the revised version of the manuscript.*

**Comment from Referee#2:** page 13. It might be worth not abbreviating in all cases as it makes it more difficult to read. For example MB perhaps just write mean bias. Similarly with SS = skill score.

*Author's response: The revised version of the manuscript will be modified accordingly.*

**Comment from Referee#2:**

- page 13. Line 13-14 "capable of significantly reducing this bias (error)". Perhaps remove the error?
- page 13. Line 32. Remove "can"
- page 15. Line 18. Typo "indicating"
- page 16. Line 2 "of" -> "by"

*Author's response: Thank, the revised version of the manuscript will be modified accordingly.*

**Comment from Referee#2:** page 16. Line 10. Do you plan to use this Baysesian method?

*Author's response: We think that several aspects of the hybrid approach are still empirical and surely a crucial aspect is to find a robust theory for an objective estimation of the relative weights. Both the methods proposed by Dobricic et al. (2015) or Menetrier and Auligne' (2015) are valid and future investigations will try to address this issue.*

**Comment from Referee#2:**

- page 16. References - inconsistent "Mon. Wea. Rev."
- page 17. Reference Mirouze has a typo "LOCKLEY" in the author list.

*Author's response: Thank, the revised version of the manuscript will be modified accordingly.*

**Comment from Referee#2:** Table 3. Why is the 0-50 mean bias worse in the bias corrected runs?

*Author's response: We think that at these depths the assumptions done to compute the bias correction are probably not adequate as explained in the manuscript (page 13 line 15).*

*Author's changes in manuscript: In the revised version of the manuscript we will expand the statements as follows:*

*"However, the systematic error correction increases the temperature mean bias between 20 and 70m depth, meaning that scales (both spatial and temporal), procedure or observation sampling used are probably not adequate at these depths. On the other hand, at similar depths, the systematic error correction reduces the salinity mean bias (Fig.9 B). We argue that temperature and salinity systematic errors in these layers have different length scales."*

**Comment from Referee#2:** Table 3. Not enough significant figures to see anything useful. I think it would more useful to use MB and SDE rather than the squares. Either that or add a significant figure.

*Author's response: We thanks the Reviewers for his/her comments. We substituted the values in Table 3 with MB and SDE not squared and not normalized. We think that together with the new figs.8 and 9 this better illustrate our results.*

*Author's changes in manuscript: New Table 3, new Figures 8 and 9 will be used in the revised version of the manuscript.*

**Comment from Referee#2:** Figure 1. Could say the date range over which the observations are plotted.

*Author's response: Thanks the info will be Included in the figure caption.*

**Comment from Referee#2:** Figure 2. Would be good to add a legend to this plot and some axis labels.

*Author's response: We thank the Reviewer.*

*Author's changes in manuscript: New Figure 2 with legend will be used.*

**Comment from Referee#2:** Figure 3. Don't understand the explanation of the middle panel. Label x axis

*Author's response: Xlabel axis will be included in the new version of the manuscript and the following caption with a better explanation of the middle panel:*

***Author's changes in manuscript:*** *New figure 3 Caption as follows:*

*"Figure 3: Example of perturbed CTD vertical profile with different quality check procedure and filtering applied. The solid black line indicates the full resolution CTD profile while horizontal lines are the associated observational error. The other colours indicate the perturbed profile. In the middle panel the 3 tested couples of horizontal correlation length scales ($L_{x,y}^{\varepsilon}$) are shown. The circles indicate the distance where the horizontal correlation of a single observation is zero. The green circle length scales are $L_x^{\varepsilon} = 12$ and $L_y^{\varepsilon} = 21$ km, this set has been used also in the reference experiment. The red circle radii are $L_x^{\varepsilon} = 6$ and $L_y^{\varepsilon} = 12$ km. The blues circle radii are $L_x^{\varepsilon} = 3$ and $L_y^{\varepsilon} = 6$ km."*

**Comment from Referee#2:** Figure 5. Spread == standard deviation of ensemble?

***Author's response:*** *Thanks, the figure caption will be modified accordingly.*

**Comment from Referee#2:** Figure 6. Are both matrices are for the same location?

***Author's response:*** *The climatological* **B** *is homogeneous and thus the same vertical correlations are applied in all the locations.*

***Author's changes in manuscript:*** *We think that changes in the manuscript due to other comments will better clarify this issue.*

**Comment from Referee#2:** Figure 7. Give the depths of the horizontal slices. Typos in the caption.

***Author's response:*** *Thanks we corrected the figure and figure caption accordingly.*

***Author's changes in manuscript:*** *New Figure 7 and new caption will be included in the revised version of the manuscript.*

**Comment from Referee#2:** Figure 8 (also Figure 9). Consider plotting MB and SDE rather than the squares it will make it easier to distinguish the lines particularly where the errors are lower. A legend would be useful on this plot too.

***Author's response:*** *We thank the Reviewer for his/her suggestion. We have re-drawn the Figs.8 and 9 using MB and SDE instead of their squares and we think Figures are improved, we also included a legend.*

***Author's changes in manuscript:*** *New Figures 8 and 9 with new caption will substitute the original ones.*

**Comment from Referee#2:** Figure 8 (also Figure 9). I notice that the non-hybrid results are quite good sometimes better than the hybrid results why might this be? It may be a case of needing to do more tuning perhaps and that not making it worse is quite a good result perhaps. It might be worth saying a bit more about this in the text.

*Author's response: The Reviewer is right, and we think that this is due to some assumptions done in our hybrid formulation. In particular the relative **B** weights and the ensemble size can be significantly improved, introducing temporal and spatial dependencies in the weight or increasing the ensemble size. However the quality of the results obtained with the hybrid scheme is, in average, better than the corresponding static formulation.*

*Author's changes in manuscript:  We will include the following statement in the conclusion section:*

[revised manuscript text omitted]

$$\mathbf{V} = \mathbf{V}_{\mathrm{D}}\mathbf{V}_{\mathrm{u,v}}\mathbf{V}_{\eta}\mathbf{V}_{\mathrm{H}}\mathbf{V}_{\mathrm{V}}. \tag{4}$$

This also has the advantage of imposing pre-conditioning, as the minimization is performed on the control variable $v$ (with $\delta x = \mathbf{V}v$), which also serves the purpose of avoiding the inversion of $\mathbf{B}$.

Basically, the background error covariance matrix is modeled as a linear sequence of several $\mathbf{V}$ operators. Each $\mathbf{V}$ defines a specific error space. From right to left $\mathbf{V}_v$ defines the vertical covariance computed using multivariate Empirical Orthogonal Functions, $\mathbf{V}_{\mathrm{H}}$ projects the vertical error to the horizontal space by means of a recursive filter, $\mathbf{V}_{\eta}$ (the balance operator) is a 2D barotropic model accounting for sea surface height adjustments and $\mathbf{V}_{\mathrm{u,v}}$ force a geostrophic balance between temperature, salinity and the velocity components. Finally, $\mathbf{V}_{\mathrm{D}}$ is a divergence damping operator avoiding spurious currents close to the coast in the presence of complex coast lines (details in Dobricic and Pinardi 2008). It is clear that this $\mathbf{B}$ formulation introduces flexibility in the code and allows the possibility to test different hypotheses.

In our static formulation of the 3DVAR, the vertical transformation operator $\mathbf{V}_v$ has the form:

$$V_V = S_c \Lambda_c^{1/2} \tag{5}$$

where columns of $S_c$ contain multivariate eigenvectors and $\Lambda_c$ is a diagonal matrix with eigenvalues of EOFs. Promising recently published results (Dobricic et al. 2015) propose a new method to estimate the vertical part of the background-error covariance matrix for an ocean variational data assimilation system based on high frequency estimates from a Bayesian Hierarchical Model. A general approach in defining hybrid assimilation schemes is to compute $B$ as a linear combination of the "static" covariance operator, $B_c$, and the flow-dependent operator, $B_e$, derived from the statistics of an ensemble of analyses/forecast:

$$B = \alpha B_c + (1 - \alpha) B_e \tag{6}$$

The relative weighting ($\alpha$) still requires empirical tuning but in general can be adjusted to the size of the ensemble. Large ensemble size can provide robust estimate of $B_e$ and thus Eq. (6) can be theoretically implemented with small $\alpha$ values (Menetrier and Auligne', 2015).

The proposed approach introduces the flow-dependent $B$ by defining the increment as a weighted sum of parts corresponding to climatological and ensemble covariance matrices:

$$\delta x = \delta x_c + \delta x_e. \tag{7}$$

It can be demonstrated (see Appendix for details) by combining Eq. (2), Eq. (6) and Eq. (7) that the following cost function has the minimum for the same value of $\delta x$ as the cost function with the background-error covariance matrix defined in Eq. (6) ( e.g. Wang et al. 2007):

$$J(\delta x) = \frac{1}{2} \delta x_c^T (\alpha B_c)^{-1} \delta x_c + \frac{1}{2} \delta x_e^T ((1 - \alpha) B_e)^{-1} \delta x_e + \frac{1}{2} (H \delta x - d)^T R^{-1} (H \delta x - d) \tag{8}$$

In Appendix we further demonstrate that by updating each member of the forecast ensemble by Eq. (8) we obtain the same estimate for the analysis error covariance matrix as when doing it with Eq. (6).

By defining the control vector $v$ consisting of climatological and ensemble parts $v = (v_c, v_e)$ the cost function becomes:

$$J(v) = \frac{1}{2} v_c^T v_c + \frac{1}{2} v_e^T v_e + \frac{1}{2} (H \delta x - d)^T R^{-1} (H \delta x - d), \tag{9}$$

and increment $\delta x$:

$$\delta x = \left(\mathbf{V}_D\mathbf{V}_{u,v}\mathbf{V}_\eta\mathbf{V}_H\right)\left(\boldsymbol{\alpha}^{1/2}\mathbf{S}_c\boldsymbol{\Lambda}_c^{1/2}\boldsymbol{v}_c + (\mathbf{1}-\boldsymbol{\alpha})^{1/2}\mathbf{S}_e\boldsymbol{\Lambda}_e^{1/2}\boldsymbol{
[revised manuscript text omitted]

[Figure]

**Figure 9: As Figure 8 but for Salinity.**

---

## Author Response (AR2)

Dear Dr. Chiggiato,

We wish to thank you for carefully reading the manuscript and your suggestion on how to improve Figure.1. In the new version of the manuscript submitted we have included a new Figure.1 with subpanels showing the entire Western Mediterranean Sea and a zoom in the area of the experiment.

5   Best regards

[revised manuscript text omitted]